# Quantifying the shift of public export finance from fossil fuels to renewable energy

Philipp Censkowsky [1,2], Paul Waidelich [3] ✉, Igor Shishlov[2,4] & Bjarne Steffen [3]

By providing guarantees and direct lending, public export credit agencies (ECAs) de-risk and thus enable energy projects worldwide. Despite their importance for global greenhouse gas emission pathways, a systematic assessment of ECAs' role and financing patterns in the low-carbon energy transition is still needed. Using commercial transaction data, here we analyze 921 energy deals backed by ECAs from 31 OECD and non-OECD countries (excluding Canada) between 2013 and 2023. We find that while the share of renewables in global ECA energy commitments rose substantially between 2013 and 2023, ECAs remain heavily involved in the fossil fuel sector, with support varying substantially across technologies, value chain stages, and countries. Portfolio 'greening' is primarily driven by members of the E3F climate club, impacting deal financing structures and shifting finance flows towards high-income countries. Our results call for reconsidering ECA mandates and strengthening international climate-related cooperation in export finance.

The increasing urgency to reduce $CO_2$ emissions requires rapidly re-directing financial flows away from fossil fuels and toward renewable energy technologies (RETs)[1–4]. To reach the required scale and speed of the transition[5,6], public financial institutions are expected to play an important role. In this light, academic research and civil society actors have scrutinized the role and financing patterns of multilateral development banks[7–9], bilateral development banks[10–12], and national state investment banks[13–15]. In contrast, public export credit agencies and export-import banks (hereafter collectively referred to as ECAs) received much less attention despite commitment volumes of ECAs being in the same order of magnitude as MDBs[8,16]. It is therefore an important empirical question whether and to what extent ECAs have re-directed their finance flows from fossil fuels to RETs. Extant research that discusses ECAs in the context of international climate finance commitments[17,18] or from a perspective of international political economy[19–22] has not conclusively answered this question (see review of extant literature and data sources in Supplementary Table 3 and 4).

ECAs support national exporters by issuing state-backed guarantees, thereby de-risking deals in overseas markets and mitigating repayment risks for private financial organizations. Outside of Europe, many ECAs also provide direct lending to deal participants[23,24]. For more than a century, ECAs played an important role in boosting domestic industrialization by ensuring international competitiveness, often tied to commercial and geopolitical goals[19,23]. On the one hand, ECAs thereby played a decisive role in the roll-out of fossil fuel infrastructure worldwide[16], with their involvement in de-risking large-scale fossil fuel projects often being criticized by civil society actors[25,26]. On the other hand, ECAs also have the potential to enhance the international financing of capital-intensive renewable energy assets[17,27,28]. In addition, international commitments have been made concerning the 'greening' of ECA activities, including the OECD ban on unabated coal-fired electricity generation[29,30] and a pledge to end overseas fossil fuel financing after 2022 by 40+ countries and financial institutions under the Glasgow Statement[31]. However, it is unclear whether these efforts to green ECAs have been

[1]Department of Strategy, Globalization and Society, University of Lausanne (HEC Lausanne), Lausanne, Switzerland. [2]Perspectives Climate Research gGmbH, Freiburg, Germany. [3]Climate Finance and Policy Group, ETH Zurich, Zurich, Switzerland. [4]Climate & Earth Center, Sustainability & Organizations Institute, HEC Paris, Paris, France. ✉e-mail: paul.waidelich@gess.ethz.ch

effective and how ECA energy finance evolves in changing geopolitical environments[32].

With this paper, we seek to fill in these gaps, drawing on commercial transaction data in export finance provided by TXF Limited, a private intelligence provider, that was previously unavailable for research. Using this data source, we identify a total of 921 energy deals between 2013 and 2023 in which ECAs from 31 OECD and non-OECD countries acted as either guarantors or direct lenders. We exclude Canadian ECA energy finance from our analysis because it is substantially underreported in the TXF data and because the Canadian ECA acts more like a state investment bank, with most ECA finance flowing to projects in Canada rather than abroad[33]. For more details on our data source and coverage, see "Methods" and the Supplementary Information. Based on transaction-level information, deal descriptions and additional desk research, we classify transactions into sectors and value chain stages following the approach of the Export Finance for Future (E3F) initiative, a climate club of ten European countries aimed to align public export finance with the objectives of the Paris Agreement[34]. By analyzing financial commitments by ECAs across our global sample of energy-related transactions, we contribute to the extant literature in three ways. First, our paper provides an academic quantification of ECA energy finance flows with a global scope that covers long-term trends over the past decade. Second, we examine deal- and tranche-level information that allows us to determine the financing patterns of ECA countries by instrument, type of technology, and deal structures, among other characteristics. Third, we show the geographic implications of shifting energy portfolios, particularly for lower-income countries, and discuss resulting equity issues.

## Results

### Aggregate trends

In analyzing financial commitments by ECAs, we distinguish two financing instruments – guarantees versus direct lending. ECA guarantees fulfill various functions but typically back commercial loans (e.g., by providing diverse risk insurances for exporters of capital goods such as ships or gas turbines). Most European 'pure cover' ECAs exclusively offer such guarantees, while others, like the North American, Japanese, or Korean ECAs, also directly extend loans (e.g., to foreign buyers of capital goods). The financing instruments of ECAs are determined by their mandate and the national public financial architecture. For instance, while European countries tend to separate state investment banks and pure-cover ECAs (e.g., in Germany, where KfW is the state investment bank and Euler Hermes is the publicly mandated ECA), ECAs in the United States or Japan combine guarantees and direct lending. We account for this heterogeneity by analyzing guarantees and direct lending separately when assessing overall global trends over four periods (Fig. 1). We combine the two as overall financial commitments in the remainder of this article and discuss relevant differences in the text where they arise.

Overall, we find that between 2013 and 2023, total ECA commitments to the energy sector varied between $USD_{2020}$ 22 and 43 billion per year, with the majority of commitments in the form of guarantees (Fig. 1a, b). Especially during P1 (2013-2015), TXF very likely underreports real total ECA energy commitments as we show in Supplementary Fig. 1, 3 and 7. On average, ECAs supported fossil energy deals with guarantees amounting to $USD_{2020}$ 15.1 billion per year and direct lending amounting to $USD_{2020}$ 9.1 billion. By contrast, annual averages for RETs amounted to only $USD_{2020}$ 4.9 billion for guarantees and $USD_{2020}$ 1.9 billion for direct lending. Since overall export finance activity plummeted during the COVID-19 Pandemic[35], a sharp decline in fossil ECA energy financing can be observed in 2021, which, for direct lending, continued in 2022 and 2023. By contrast, RET commitments have remained relatively stable and, in 2023, increased to $USD_{2020}$ 9.9 billion and $USD_{2020}$ 4.4 billion for guarantees and direct lending, respectively. For the first time, this jointly exceeded fossil-related

commitments that stood at $USD_{2020}$ 11.4 billion and $USD_{2020}$ 2.6 billion, respectively. As a result, the share of ECA energy commitments to RETs has markedly increased from 9% and 5% in 2013 to a two-year average of 42% and 39% for guarantees and direct lending in 2022-2023, respectively (Fig. 1c). This shift from fossil energy towards RETs is reflective of global investment patterns in the energy sector[36]. Interestingly, the Pandemic-related uptick in RET shares occurred two years earlier for ECAs' guarantees compared to direct lending, primarily due to an increase of RET financing by European pure-cover ECAs that do not engage in direct lending.

### Trends along the fossil fuel value chain and by technologies

We disaggregate ECA involvement along different stages of the fossil fuel value chains and across renewable power generation technologies in Fig. 2. By considering the annual average commitments of each period, several key trends are observable. First, after the 26th UN climate conference (COP26) in Glasgow in 2021, ECAs in the OECD have ceased their international coal power financing following a ban on export finance for coal-fired electricity[29]. Importantly, ECAs of non-OECD countries also exhibit a concurrent reduction in coal financing, except for one Chinese guarantee for a new 1320 MW coal power plant in Pakistan in 2023. Second, in oil value chains, upstream transactions, such as oil platforms or drilling vessel acquisitions, dominate together with support for downstream activities, such as refineries. The large drop in downstream oil financing during the Pandemic (P3) was partially compensated by twelve larger-scale deals involving refineries in Indonesia and Egypt in the post-Glasgow period (P4). Third, while nearly on par with oil value chains through 2019, deals involving natural gas received the highest levels of ECA commitments during the Pandemic (P3), with annual financing volumes reaching an average of $USD_{2020}$ 14.3 billion in 2020-2021. The subsequent, massive drop in gas financing in the post-Glasgow period (P4) is potentially due to post-pandemic risk aversion for long-term commitments in supporting larger-scale upstream or downstream projects[36]. An additional explanation is that large untied state guarantees to secure foreign gas supplies (e.g., in the case of Germany[37]) are not included in our transaction data.

The sharp fluctuations in ECA support for fossil fuels over the recent years contrast with the steadily growing RET commitments. Indeed, in the post-Glasgow period (P4), ECA commitments for RETs are for the first time higher than for fossil fuel commitments combined, with wind projects representing about two-thirds of total RET commitments. This trend is primarily linked to ECA support for large offshore wind projects that tend to be riskier[38,39], such as the Doggerbank wind park in the North Sea that will become the largest offshore wind park worldwide, supported by ECAs from Norway, Sweden, and France. In addition, in 2023, a USD 8 billion deal was partially covered by the German ECA Euler Hermes, leading to the first ECA-supported large-scale green hydrogen and ammonia facility that will be built in Saudi Arabia. In the Supplementary Information, we further report additional RET financing in P1 and P2, especially from China (Supplementary Figs. 4 and 6b). Overall, total yearly commitments for fossil and renewable power generation at 12.9 $USD_{2020}$ over the period 2013 to 2023 place ECAs in the same order of magnitude as multilateral development banks[8], highlighting the key role of ECAs in energy finance worldwide.

### Trends by countries and climate club membership

While highlighting the overall trends of ECA finance flows, globally aggregated numbers potentially mask distinct trends concerning different geographies. Accordingly, we further disaggregate financing patterns by country and country blocks to illustrate heterogeneities depending on their membership in the E3F climate club (Fig. 3). Launched in 2021, the E3F coalition consists of 10 European countries that pledged to align their export financing portfolios with the Paris

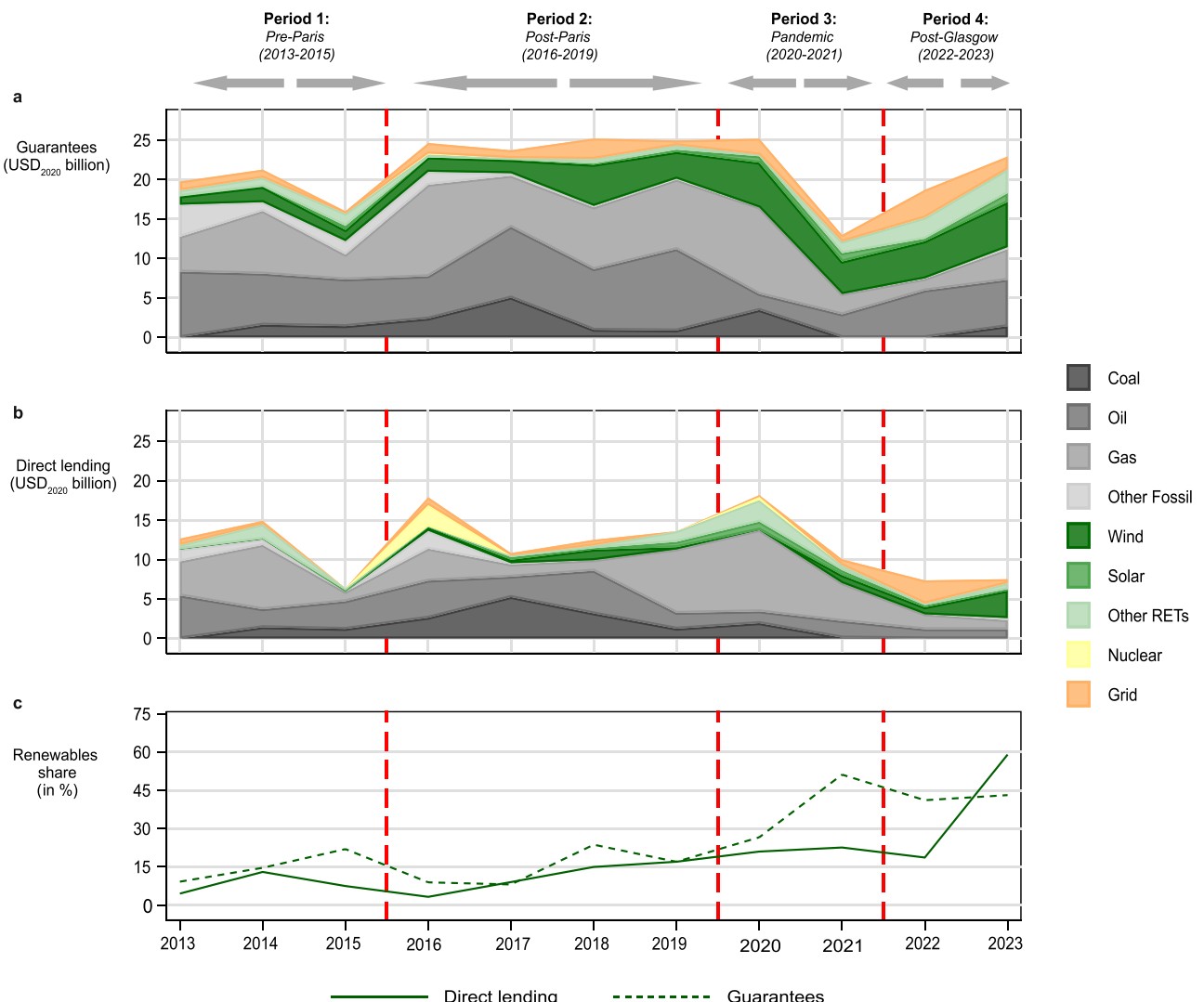

**Fig. 1 | Aggregate trends of energy finance by export credit agencies. a** Global sum of ECA energy guarantees by year. **b** Global sum of ECA direct lending to energy projects by year. Area colors indicate different energy subsectors across fossil fuel, renewables, nuclear and grid projects. **c** Share of RETs over total ECA energy lending and guarantees, respectively. Data coverage is global except Export Development Canada, the Canadian ECA (see "Methods" and Supplementary Fig. 7). Note that some countries do not fully report transactions to TXF (e.g., China and Mexico, see Supplementary Fig. 1, 2b). We triangulate the results displayed here using tertiary data from Oil Change International, a non-governmental organization, in Supplementary Fig. 3. We separately report the tertiary data for Canada in Supplementary Fig. 7. 'Other fossil' includes deals in the power ($n = 44$) and shipping sector ($n = 6$) that could not unambiguously attributed to one type of energy (e.g., conventional/mixed power plants or ships used for both oil and gas extraction) as well as conventional hydrogen ($n = 1$). 'Other RETs' includes mixed renewables ($n = 50$), hydro ($n = 25$), waste-to-energy ($n = 8$), biogas ($n = 6$), biomass ($n = 6$), geothermal ($n = 5$) and green hydrogen ($n = 3$).

Agreement[34,40]. Specifically, this includes to "end new direct public support for the international unabated fossil fuel energy sector, except in limited and clearly defined circumstances in line with the 1.5° C scenario, by the end of 2022"[41]. In addition, E3F members now seek to increase and report on their ECA climate finance contributions, such as the new collective quantified goal of climate finance negotiated at COP29 in Azerbaijan[42]. As a result, the E3F coalition is the only group of ECA countries that reports consistently on their energy finance and has a clear focus on policymaking in export finance. By contrast, some ECAs and their guardian authorities congregate in other climate initiatives in the sector[43], such as the Net-Zero Export Credit Agencies Alliance[44] or the Clean Energy Transition Partnership, an initiative following the Glasgow Statement at COP26[31]. We did not choose these initiatives as grouping categories since the former is more practitioner-oriented, and the latter is a broader public finance initiative that includes non-ECA organizations.

Our results support that the E3F coalition has a higher share of commitments going to RETs and grid projects and has 'greened' their portfolios more decisively over the past decade (Fig. 3a). As fossil-related commitments shrink globally, the E3F coalition gains relative weight after the Paris Agreement and particularly in the post-Glasgow period, thus driving the overall shift from fossil fuels to RETs in aggregate ECA financing. In non-E3F countries (Fig. 3b), Japan shows a clear tendency of portfolio 'greening' post-Glasgow (P4), which is, however, accompanied by significantly lower average annual energy commitments than before the Pandemic (−50% comparing annual averages between P1-P2 with P3-P4) which reflects the post-pandemic contraction of the Japanese economy[45]. In South Korea, fossil fuels still account for 71% of 2022-2023 energy commitments, driven by high ECA support for exporting LNG tankers from Korean shipyards[46]. Non-OECD countries, incl. China, which accounts for 73% of total commitments of this subsample, exhibit a return to fossil fuel deals since the

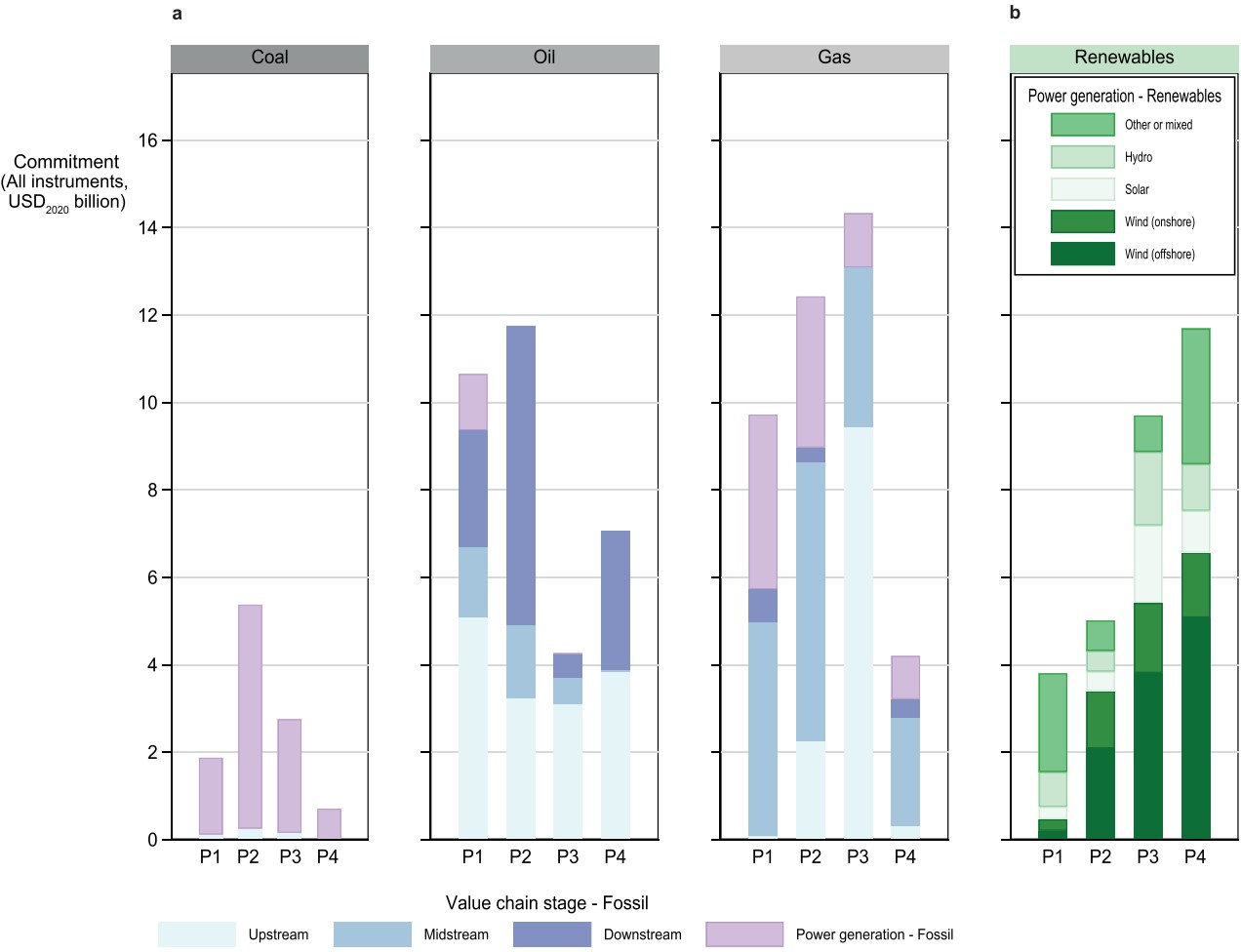

**Fig. 2 | Average annual commitments by value chain stage (fossil fuels) or power generation technology (RETs). a** Average annual commitments in fossil fuel value chains. **b** Average annual commitments for renewable power generation projects. Commitments aggregate both direct lending and guarantees and are calculated within periods. The definitions of each value chain stage or technology correspond to available ISO standards (see "Methods"). We omit nuclear projects here, given the small number of deals in our sample ($n = 4$). The category 'Other or mixed' for Renewables includes deals in mixed or unspecified renewables ($n = 95$), waste-to-energy ($n = 8$), biogas ($n = 6$), biomass ($n = 6$), geothermal ($n = 5$) and green hydrogen ($n = 3$). Data coverage is global but excludes Export Development Canada, the Canadian ECA, and underrepresents some countries that insufficiently report to TXF according to data triangulation (e.g., China and Mexico, see Supplementary Fig. 1, 2b). We triangulate the results displayed here using tertiary data from Oil Change International, a non-governmental organization, in Supplementary Fig. 4. Period P1 refers to Pre-Paris (2013-2015), P2 to post-Paris (2016-2019), P3 to the Pandemic (2020-2021), and P4 to post-Glasgow (2022-2023).

Pandemic. Meanwhile, non-E3F countries from the OECD show a slow but constant transition away from fossil fuels. Similarly, most E3F member countries are fully shifting away from fossil fuels (Fig. 3c) as they implement exclusion policies aligned with commitments made under the Paris Agreement and the Glasgow Statement[34]. The increasing weight of the E3F coalition reflects a more decisive stance of the European Union in the ongoing OECD negotiations around tighter oil and gas restrictions[47]. Among E3F members, Italy is a notable exception with continued high levels of fossil fuel support, while Denmark has been leading with 100% support for RETs and grid infrastructure, reflecting its strong domestic wind energy industry. Concerning grid infrastructure, which is required for integrating more RETs in the electricity systems, Germany's ECA Euler Hermes is leading with 36% of total energy commitments in 2022–2023 supporting the sector, including a state-of-the-art battery cells factory in Europe and rural electricity grid expansion in Angola.

Notable is the absence of U.S. EXIM Bank and Export Development Canada, the Canadian ECA, among the largest ECAs supporting energy infrastructure worldwide. The absence of EXIM is explained by internal governance challenges (a board quorum lapse) between 2015 and 2019, which prevented the agency from financing deals above USD 10 million[48]. After the quorum was reestablished, however, the U.S. EXIM Bank resumed supporting oil value chains even in the post-Glasgow period and has more plans in the sector, despite the U.S. commitment to end overseas fossil fuel financing[49,50]. By contrast, because of its unusually high domestic loan portfolio[16,33], the Canadian ECA reports incompletely to TXF which is why we excluded the country from the results presented here (see "Methods" as well as Supplementary Fig. 7 for a detailed complementary analysis).

## Deal financing characteristics by technology

To explore how the overall shift in ECAs' energy finance commitments towards RET will likely impact deal financing structures, we break down deal characteristics by technology. Figure 4a illustrates the deal size distribution by technology with additional labels for outlier mega-projects, such as the Yamal LNG project in Russia's Arctic, the Mozambique LNG project, or the Neuconnect submarine power cable that is currently under construction between the UK and Germany. The median deal size is the largest for oil and gas projects with USD$_{2020}$ 492 million and USD$_{2020}$ 289 million, respectively. This compares to USD$_{2020}$ 133

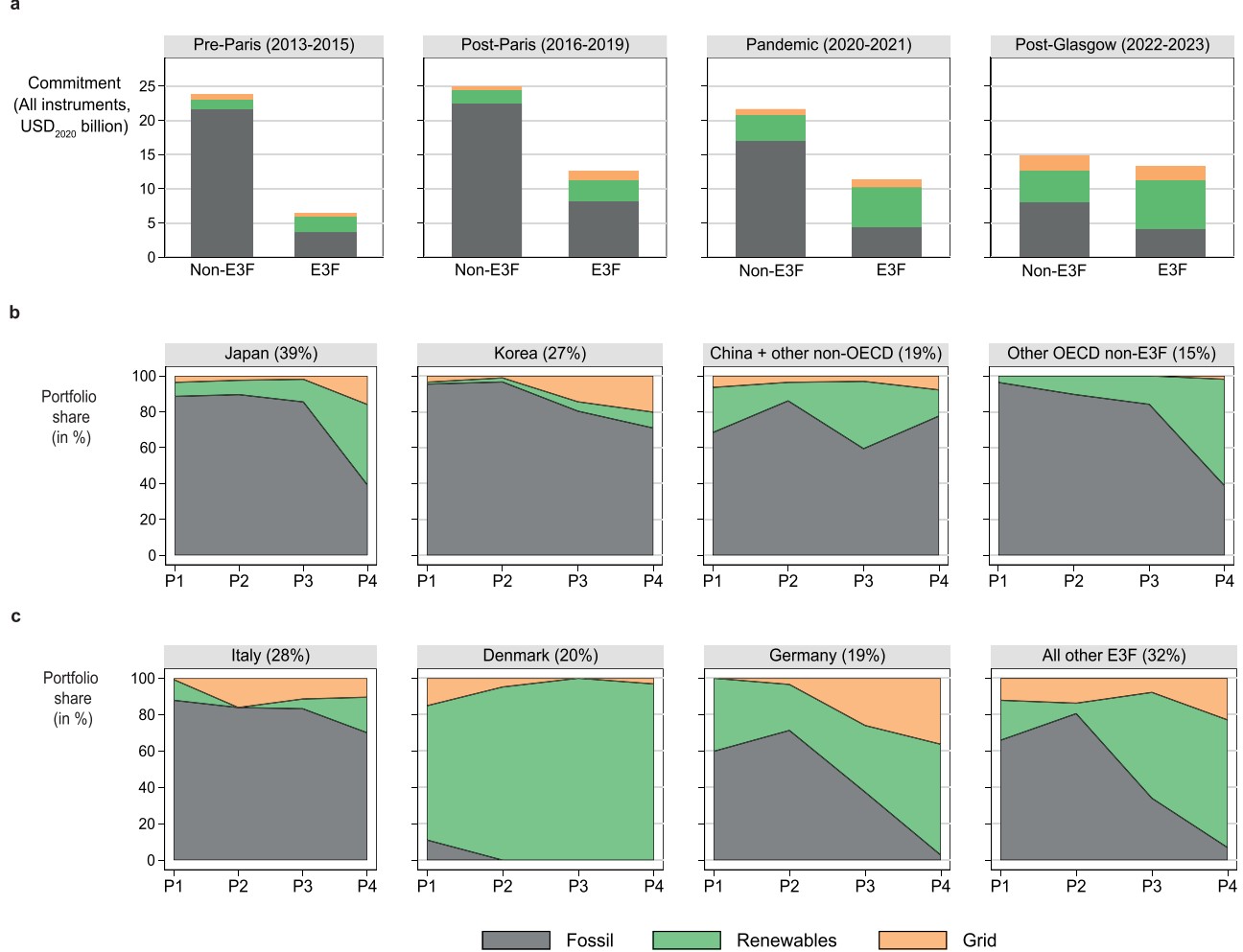

**Fig. 3 | Absolute and relative portfolio 'greening' trends within and outside the E3F member countries.** **a** Comparison of average annual energy commitments between non-E3F versus E3F country groups by period. **b** Share of commitments for fossil fuel projects, RETs, and grid-related projects (incl. storage) in non-E3F countries. **c** Share of commitments for fossil fuel projects, RETs, and grid-related projects (incl. storage) in countries of the E3F climate coalition. Countries are ordered by their total cumulative energy finance commitments within each group. E3F member countries include Belgium, Denmark, Finland, France, Germany, Italy, Netherlands, Spain, Sweden, United Kingdom. Non-E3F countries are the remaining 22 countries in the TXF dataset, out of which nine are non-OECD countries (listed in Supplementary Table 1). Percentage labels in parentheses indicate the share of a country's cumulative energy finance commitments between 2013-2023 within each group (E3F or non-E3F). We omit nuclear projects here, given the small number of deals in our sample ($n = 4$). Data coverage is global but excludes Export Development Canada, the Canadian ECA, and underrepresents some countries that insufficiently report to TXF according to data triangulation (e.g., China and Mexico, see Supplementary Fig. 1–2b). We triangulate the results displayed here using tertiary data from Oil Change International, a non-governmental organization, in Supplementary Fig. 5. Period P1 refers to Pre-Paris (2013–2015), P2 to post-Paris (2016–2019), P3 to the Pandemic (2020–2021), and P4 to post-Glasgow (2022–2023).

million, USD$_{2020}$ 55 million, and USD$_{2020}$ 113 million for wind, solar, and grid projects, respectively. Hence, as ECAs are shifting towards renewables, deals are becoming on average two to three times smaller.

Figure 4b displays the loan tenor by technology, i.e., the maximum repayment terms of a loan. This analysis is policy-relevant since increasing the maximum tenors to up to 22 years for climate-related projects was a central element of a recent modernization of the OECD Arrangement, the main supranational regulatory framework for officially supported export credits (see Supplementary Note 1 for a more detailed description)[51,52]. Indeed, we find that the median tenor of wind energy projects and other non-solar RETs, such as hydro power, is higher than for any other technology. The longest tenor in our dataset is 40 years, granted by several multilateral banks, a Korean ECA, and the Green Climate Fund to the 15 MW Tina River Hydropower project on the Solomon Islands. However, we find no evidence that non-OECD countries—that are not bound by the Arrangement—would offer more flexible financing conditions for maximum tenors than their OECD counterparts.

On the contrary, the average tenors supported by ECAs within the OECD Arrangement are two years higher than those by non-OECD ECAs.

Furthermore, we find that the most frequent borrower types involved in ECA-supported deals are special purpose vehicles (Fig. 4c), i.e., stand-alone entities commonly used by deal participants to isolate risks in project finance[53]. The share of special purpose vehicles is particularly high for wind energy projects and particularly low for grid projects where government entities account for more than 50% of borrowers, given the central role of state ownership in power grids[53,54]. Finally, against initial expectations, the type of financing instrument does not vary strongly across technologies (Fig. 4d). In ECA-supported deals, about 60–90% of financing instruments are either fully (or partially) covered or directly extended by an ECA.

### Geographic implications of shifting energy portfolios
Relevant to a comprehensive understanding of ECAs' role in the energy transition are not only the trends in technologies supported by public

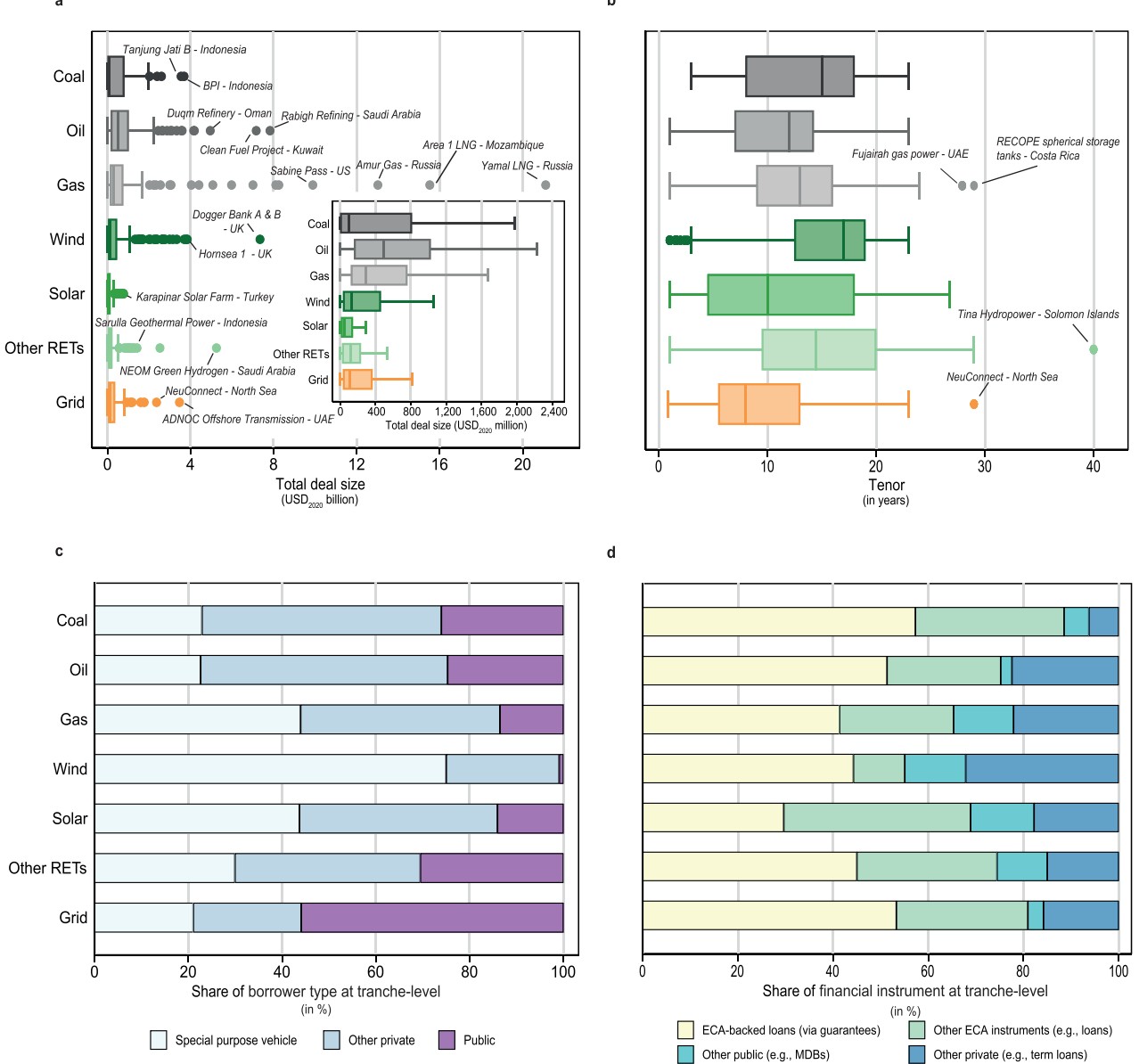

**Fig. 4 | Deal financing characteristics by technology. a** Total deal sizes of ECA-supported energy projects ($n = 866$). Boxes indicate the median and the 25% and 75% of the deal size distribution. Whiskers extend to values of 1.5 times the inter-quartile range from the upper (or lower) quartile. The largest outliers are labeled for each technology, and we include a zoom-in showing the boxplots on a smaller $x$-axis scale. **b** Tenors of loan tranches in ECA-supported deals ($n = 1768$). Boxplots are equally defined as above. **c** Share of borrower types for tranches in ECA-supported deals. 'Other private' includes private companies, listed companies, financial institutions and investment managers, while 'Public' includes government-owned companies, governments, ECAs and public-private partnerships. Note that we omit one unspecified borrower category 'Other' in Panel C ($n = 5$ tranches). Shares are

calculated by dividing the count per category by the total number of tranches. **d** Share of financial instruments in ECA-supported deal tranches. 'Other ECA instruments' refers to direct loans (75%), supplier credits (16%), performance bonds (7%), letter of credits (2%) and Islamic finance (< 1%). 'Other public' refers to direct loans (93%) or guarantees (7%) by multilateral development banks or bilateral development institutions. 'Other private' refers to private term loans (84%), revolving credit facilities (3%), value-added tax facilities (3%), commercial loans (2%), and a dozen other small instrument categories (each less than 1%). We omit nuclear projects because of the small sample size ($n = 4$) as well as fossil-related deals from the power ($n = 44$), shipping ($n = 6$) and conventional hydrogen ($n = 1$) sectors that cannot be attributed to a succinct category.

funds but also their allocation among recipient countries. Figure 5a shows the geographic distribution of recipients of ECA energy commitments between 2013–2023 for fossil fuel and RET projects. While ECA support for fossil fuels primarily flows into leading oil and gas producing countries, such as the United States or Russia, commitments for RETs are more concentrated in Europe, with Angola being a notable exception due to two large hydro and two larger solar PV deals supported by European ECAs. Indonesia hosts many larger-scale oil refinery and coal- or gas-fired power projects, many of which are supported by Japanese or Korean ECAs. As discussed above, European

countries are an increasingly prominent provider of ECA energy finance. Therefore, compared to fossil fuel finance, ECA support for RET projects is much more likely to remain in the same region (Fig. 5b). For instance, 70% of Denmark's ECA commitments for RET between 2013-2023 have gone to energy projects located in Europe. As a result, the sharp increase of RET projects in overall ECA commitments observed during the post-pandemic period and the rising market share of the E3F coalition has come with a considerably higher share of high-income countries among the recipients of ECA energy finance at the expense of upper-middle and lower-income countries (Fig. 5c). Hence,

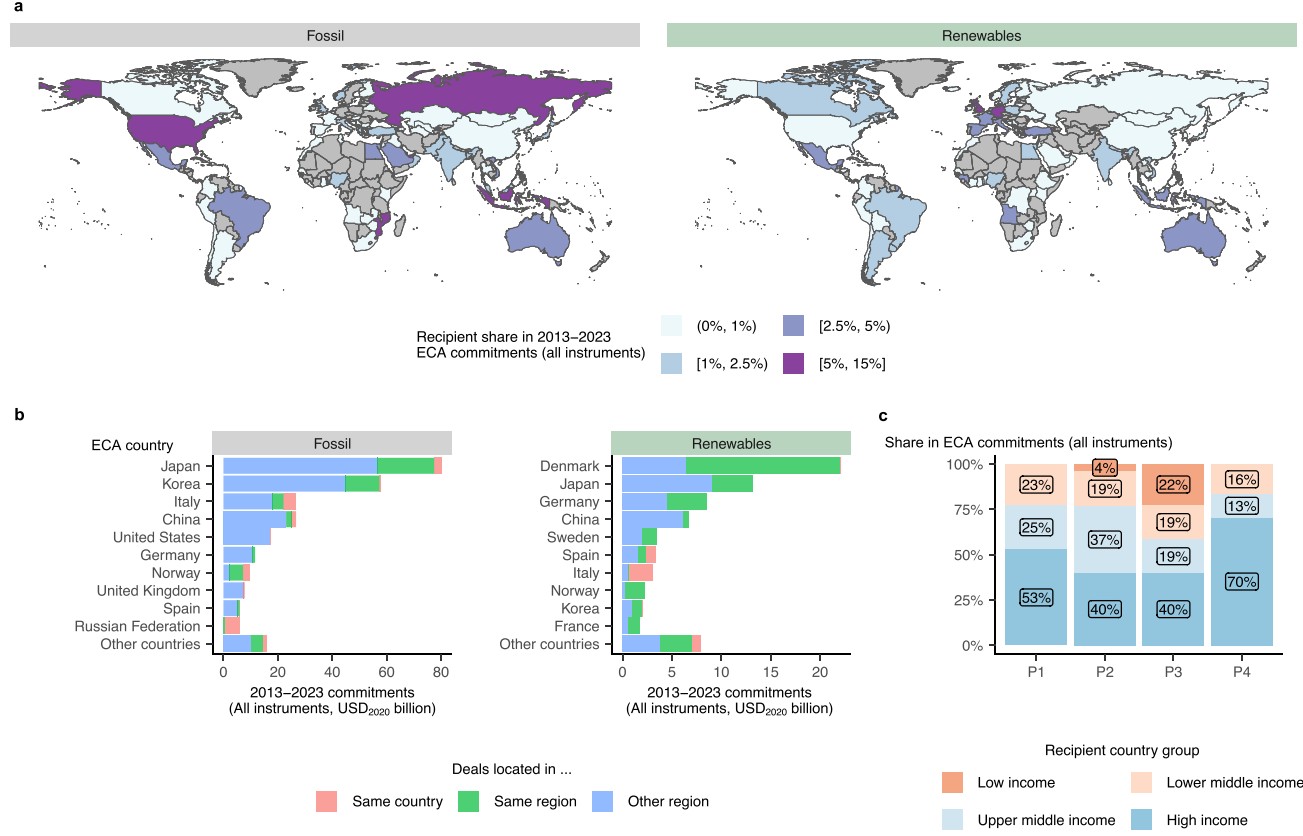

**Fig. 5 | Distribution of ECA energy finance commitments for fossil fuel and RE projects across countries, regions, and income levels. a** Share of each recipient country in the overall 2013–2023 ECA energy finance commitments, including both direct lending and guarantees for fossil fuel and RE projects. Shares are binned for the sake of readability. Natural Earth shapefiles are sourced via Andy South's rnaturalearth package[94]. **b** Overall ECA commitments by ECA country over 2013-2023, with bars indicating the share of commitments received by countries in the same country (red), same region (green), or a different region (blue). Region definitions are taken from TXF and distinguish between Africa, Asia, Asia Pacific, Australasia, Europe, Latin America, the Middle East, North America, and Russia CIS. **c** Annual share of recipient countries grouped by the World Bank's income level categorization in overall 2013–2023 ECA energy finance commitments. Data coverage is global but excludes Export Development Canada, the Canadian ECA, and underrepresents some countries that insufficiently report to TXF according to data triangulation (e.g., China and Mexico, see Supplementary Fig. 1, 2b). We triangulate the results displayed here using tertiary data from Oil Change International, a nongovernmental organization, in Supplementary Fig. 6.

current trends of 'greening' ECA portfolios are accompanied with a marked shift towards high-income countries (according to World Bank country classification, see "Methods").

The only significant exception in our sample is Mozambique, one of the world's lowest-income countries, which accounts for nearly 22% of ECA energy commitments in P3. Here, several ECAs supported the stepwise development of natural gas reserves estimated at up to 4,247 billion cubic meters[55], about twelve times Europe's 2023 annual gas demand[56]. With total commitments of USD_2020 of 13 billion for Mozambique LNG in 2020, this project alone represents about four times Mozambique's average foreign direct investment between 2018 and 2022[57]. However, escalating violence following insurgency attacks in the region led the project developers to suspend Mozambique LNG in 2021[58]. As of 2024, however, some industry observers estimate that, in 2025, the project may come back on track together with an even larger sister deal to develop adjacent fields[59].

## Discussion

Our aggregate results substantially differ from previous estimates of ECA-related RET or 'climate' financing. For example, the Climate Policy Initiative had priorly estimated ECA contributions in 2021–2022 with annual USD_2020 1.7 billion only but excluded guarantees[60], while we find a yearly average of USD_2020 10.8 billion since 2021 (USD_2020 2.7 billion of direct lending and USD_2020 8.1 billion of guarantees). However, this is still about five times lower than the minimum needs for climate-related ECA commitments estimated by another study at USD_2020 43–55 billion per year until 2030[17]. Therefore, RET financing by ECAs needs to be further scaled up to achieve the 'fair share' in international climate targets[17]. A definitive end to fossil fuel support by ECAs is not observable, and despite the sharp decrease during the Pandemic, support for oil and gas value chains via guarantees is picking up again in 2022-2023. Such continued ECA support for oil and gas chains is at odds with most countries' climate commitments, notably under the Paris Agreement and the Glasgow Statement[31]. Oil and gas exclusions are currently missing from the OECD Arrangement but are being discussed at the OECD (see Supplementary Note 1)[47]. The growing importance of RETs in the portfolios of ECAs could help to overcome countries' reluctance to introduce such restrictions and exclusions.

Outside the OECD, we find that especially Chinese ECAs, Sinosure and the Export-Import Bank of China, dominate ECA energy financing, albeit with an increasing share of fossil fuels (Fig. 3b, also see Supplementary Fig. 2b and Supplementary Table 6). Chinese ECAs today represent a major counterweight to traditional ECAs, and fears of

giving away market shares to China inhibit (climate) policy changes within the OECD, especially from regional competitors South Korea and Japan[11,19,61]. In general, ECA policies remain uncoordinated between OECD and non-OECD countries, given disagreements about transparency issues which caused the suspension of the International Working Group on Export Credits, the only transpacific forum for public export finance policies previously led by the United States and China[62]. Given the recent come-back of Chinese coal financing[63], policymakers should hence consider relaunching the Working Group for climate cooperation. In this context, the E3F coalition could play a pioneering role in demonstrating which policies can incite exporters to pivot into sustainable markets, for instance, by aligning ECAs' mandates with the Paris Agreement and offering favorable terms and conditions for Paris-aligned projects. The finding that, since 2022, joint annual energy sector commitments by E3F countries are almost on par with non-E3F countries (USD$_{2020}$ 13.3 billion versus USD$_{2020}$ 14.9 billion) underlines the growing importance of this coalition.

Beyond trends and patterns in climate-relevant ECA activities, our results highlight strong discrepancies between high-income and developing nations, with the former accounting for most ECA commitments (see Supplementary Tables 6 and 7 for a split between OECD and non-OECD country financing). Aside from China, non-OECD countries in our sample, such as Indonesia, Russia, or Saudi Arabia, founded their ECAs only in the 1990ies or 2000s and show comparatively little but growing ECA activity[64]. On the recipient side, we find that ECA portfolios markedly shift to high-income countries as the energy transition is ongoing. This is particularly concerning for the clean energy financing needs of emerging and lower-income nations, e.g., on the African continent[65,66]. More generally, the observation that ECAs rarely support deals in low-income countries can be explained by economic considerations. Unlike development finance institutions whose mission is to support socio-economic development, ECAs traditionally help national exporters to realize deals primarily based on commercial profitability. This is scarcely the case in low-income countries that typically require highly concessional conditions, e.g., from multilateral entities at the World Bank Group[67]. One way to counterbalance the geographical trends identified in our study could be to expand ECA mandates by explicitly supporting the energy transition in lower-income countries[68,69].

ECA climate policies can have important adverse implications for traditional export industries. In line with evidence showing export technologies being a major 'push factor' of financing activities[70], ECA portfolios typically reflect the composition of national export industries[41]. In most countries, this historically included goods and services related to the production, transport or use of fossil fuels, rather than RETs. However, instead of largely remaining demand-driven agencies, there are important calls for ECAs to become active agents in the pursuit of reconciling climate, industrial and trade policy objectives[32]. Indeed, some ECAs already support ventures into emerging green markets that require technical capabilities similar to those of traditional export industries[71]. In 2023, for instance, the German ECA provided cover to build the first large-scale green hydrogen and ammonia facility in Saudi Arabia[72] and the first large-scale green steel factory in Sweden[73]. Regarding employment in traditional energy export industries, input-output modeling suggests overall positive employment effects of investment in more labor-intensive RETs[74] and specifically for shifting ECA support towards RETs for some countries[75,76]. However, historical experiences of transitions away from coal highlight the economic risks, especially in the labor market[77]. To soften employment impacts, withdrawing ECA support for fossil fuel technologies could be combined with additional policies facilitating the transition of workers to cleaner exporter industries, e.g., retraining and compensation schemes[78].

Besides traditional export industries, ending ECA financing for fossil fuels can also affect host countries that are economically dependent on fossil fuel exports, such as the United Arab Emirates or Mozambique[79]. In our sample, a significant share (40%) of total ECA energy finance flows to fossil fuel-dependent host countries, mostly supporting fossil fuels rather than RETs or the grid sector (see Supplementary Note 2). Therefore, such countries may advocate for longer transition periods and continued eligibility for ECA support in the fossil fuel sector to attenuate the potentially adverse economic effects of fossil fuel phase-outs[66,79,80]. This may include oil and gas workers whose skills are not directly transferrable to other sectors, e.g., in the highly specialized and labor-intensive parts of the upstream industry[81]. Yet, besides the climate and environmental risks of new fossil fuel infrastructure[82,83], continued eligibility may also increase risks of asset stranding[80]. Policy-makers should therefore prioritize phase-out policies that sufficiently consider the project country context, thus ensuring a just transition[66,68,84].

Overall, our paper demonstrates the empirical relevance of ECAs as a previously under-researched group of public finance institutions in the energy transition. While our sample exhibits an unprecedentedly high degree of coverage regarding countries and guarantee volumes, there are important limitations to our analysis that leave gaps for future research. Most pertinently, our results exclude Export Development Canada, the Canadian ECA, and likely under-report some countries that do not completely report deals to TXF, e.g., China and Mexico (see Supplementary Fig. 2a, b). As a corollary, our results present a lower bound of ECA financing, especially concerning direct ECA lending to fossil fuel projects. To address this limitation, we conduct a robustness check in which we rely on data gathered by the nongovernmental organization Oil Change International wherever TXF deals appear missing or incompletely reported (see "Methods" and Supplementary Fig. 1–7). While the overall trends and conclusions identified in this study remain consistent, the robustness check suggests that the rise of RET share in overall ECA energy finance may be somewhat less pronounced and the magnitude of total ECA energy commitments significantly higher, particularly between 2013 and 2015. These remaining uncertainties underpin the need for public policy efforts to improve data transparency, especially concerning the transaction databases hosted by the OECD and the Berne Union, the leading global association for the export credit and investment insurance industry, both of which - much unlike data from other public finance institutions - remain inaccessible to the public.

Aside from improving data transparency, other fruitful avenues for future research remain. First, the scope of our study is restricted to public ECAs and the energy sector, while public ECAs represent only half of the Berne Union's member organizations[85]. Future research could therefore explore the role of private export credit and investment insurance agencies, as well as the role of ECAs in de-risking infrastructure in non-energy but climate-relevant sectors, such as mining, industry, and transport. Second, our study provides descriptive evidence on ECA financing patterns following policy events, such as the OECD-wide ban of ECA support for coal-fired electricity generation, but it does not identify the causal impacts of policy changes. Here, future research could use quasi-experimental designs to isolate the effects of specific policy measures from broader trends. Third, we do not examine the economy-wide second-order effects of shifting ECA portfolios towards RETs, for instance, on employment or public budgets. To fill in these gaps, case studies using qualitative methods and modeling studies could help to better understand context-dependent second-order effects and risks and opportunities for affected industries, especially in economies that depend on exports of fossil fuels or equipment used in oil and gas value chains. Fourth, our study only examines energy deals with ECA involvement, whereas future research could systematically compare deals with and without ECA involvement to identify drivers of ECA involvement (data availability currently limits our analysis in that regard)[15]. Finally, our study

adopts a backward-looking approach by analyzing closed deals. Regarding the financing needs of low-carbon transitions in lower-income countries in particular, scholars could complement our study using forward-looking methods and examine the potential role of ECAs in meeting these needs. One way could be using computational models that quantify energy system investments under different scenarios and consider ECA effects through a reduction of project-specific cost of capital (which have a markable effect on projected investment volumes especially in lower-income countries). To calibrate such forward-looking analyses, more empirical work is needed concerning the precise impact of ECA involvement on financing conditions for both fossil fuel and renewable energy projects[86–88]. The insights derived from such studies could help policymakers leverage ECAs more effectively to finance and foster the energy transition.

## Methods

### Transaction data and scope

Transaction data covering $n = 921$ deals that involve an ECA between 2013-2023 is provided by TXF Limited, a private intelligence provider that relies on self-reporting of closed deals by deal participants, including ECAs. Nearly all public ECAs confidentially report to TXF in exchange for visibility of what competitors are doing (see Supplementary Table 1 for the complete list of the 31 countries and 45 ECAs included in this study). A notable exception is Export Development Canada, the Canadian ECA, which is not included in the results presented in the main manuscript. This is significant since the ECA is among the top energy finance providers with annual energy commitments exceeding $USD_{2020}$ 8.9 billion between 2017 and 2022 according to Oil Change International, a non-governmental organization[16]. However, since up to 90% of the ECAs' energy-related commitments are granted domestically[33], the ECA therefore has no incentive to report to TXF. We use tertiary data from Oil Change International wherever TXF coverage is either missing (Canada) or likely under-reported (see 'Data triangulation' below and Supplementary Figs. 1–3b). In sum, however, TXF data features rich tranche-level information for the entirety of the energy sector (RETs, fossil and grid infrastructure) that we can leverage in this contribution. For a comparison of empirical data sources and previous peer-reviewed studies, see Supplementary Tables 3 and 4, respectively.

We define ECAs as per their primary mandate at the country level, which is more restrictive than the original set of organizations provided by TXF. Moreover, we only retain public ECAs with at least one deal in the energy sector since 2013. These conditions led us to exclude a total of 40 of the 85 financial organizations in TXF data that assume ECA-like functions (e.g., providing risk insurance via guarantees) but do not fall under the scope of frameworks such as the OECD Arrangement, are not listed as official export credit agencies by the OECD[89], or are public ECAs but display no energy sector involvement between 2013 and 2023. These organizations are listed in Supplementary Table 5, and the majority include multilateral or bilateral institutions, such as the Multilateral Investment Guarantee Agency, the Korean Development Bank, or the European Investment Bank. Removing organizations within the energy sector scope of this paper ($n = 20$) implies a reduction of the total sample size by 49 deals corresponding to a reduction of total cumulative ECA energy commitments by about 3%. We analyze ECAs at the country level (rather than the organizational level) because ECA policies typically concern all ECAs in a country. Finally, to account for the heterogeneity of ECA mandates and financing capabilities, we include ECA support via guarantees and direct lending in international as well as domestic deals. As ECAs are typically mandated with supporting exporters abroad, domestic commitments represent only about 8% of total commitments in our dataset.

### Classification of energy-related transactions

The TXF database classifies deals into 12 deal industries (e.g., 'transport') and 128 diverse deal sub-industries (e.g., 'oil – midstream transportation'). To identify all deals related to the energy sector, we applied the energy sector definition of the so-called Joint Transparency Reporting, a methodology published by the E3F climate club[34]. Given its suitable logic for defining fossil fuel value chains in their entirety (upstream, mid-stream, downstream, power generation) according to the ISO 20815 standard, we adopted the E3F approach with minor modifications for the study at hand (see Supplementary Table 2).

Since TXF does not classify deals according to this definition, we reclassified the dataset using the following procedure. First, we removed sectors with evidently no relation to the energy sector (e.g., 'Agri-/soft commodities'). We then allocated deals from obviously energy-related industry categories (e.g., 'oil & gas' or 'renewables') to their precise value chain stage or RET. For less obvious categories (such as 'infrastructure' or 'transport'), we used an algorithm-supported systematic search for keywords that was applied to the titles, borrowers, and descriptions of deals, where available. Examples of key words used to identify potentially relevant deals include terms such as 'gas', 'LNG', 'drilling', 'renewable' or 'wind'. This procedure permitted us to identify several hundreds of potentially energy-related deals. In a subsequent step, these deals were screened manually and attributed to the correct energy or technology category by two researchers (P.C. and P.W.), where applicable. In a few cases, web searches were necessary to verify information on company websites or media archives. In total, we retained 113 deals that were clearly congruent with our definition of the fossil energy sector but were priorly not classified as such (representing about $USD_{2020}$ 30.4 billion in ECA commitments). More than half of these deals involved guarantees or lending for the export of ships, such as LNG tankers or upstream drilling ships. To be conservative, we did not retain deals involving multi-purpose vehicles, such as dredging vessels that are often (but not exclusively) used to clear seabeds in offshore oil and gas production. Meanwhile, only eight deals representing $USD_{2020}$ 357 million were reclassified as renewable energy-related projects. In addition, we verified the exact nature of ambiguous transactions in the category 'Conventional power' ($n = 56$). Based on a case-by-case check and desk research, we re-classified these deals as 'Coal' ($n = 1$), 'Oil' ($n = 3$), 'Gas' ($n = 8$), 'Oil & gas mixed' ($n = 2$), 'Other fossil' ($n = 36$), 'Solar' ($n = 3$), 'Hydro' ($n = 2$), and 'grid infrastructure' ($n = 1$).

### Data triangulation

We triangulate our results with the only other larger available data source on official export finance: The Public Finance for Energy database (https://energyfinance.org/#/data)[16] maintained by the non-governmental organization Oil Change International (OCI) for 16 ECA countries (versus 31 countries in the TXF database). Other data sources, such as those by the European Joint Transparency Reporting by the E3F coalition[34], provide granular reporting only for 10 European countries, while OECD reporting[90] is highly aggregate and excludes non-OECD countries. We provide an overview of all other available data sources and prior empirical studies on ECAs in Supplementary Table 3. The data triangulation proceeded as follows. First, we compared aggregate trends of all commitments for fossil, RETs, or grid projects between OCI and TXF (Supplementary Fig. 1) and show the differences of coverage in detail for each country and instrument (Supplementary Fig. 2a for guarantees and 2b for direct lending). While OCI has higher coverage for the direct lending portfolios of ECAs in China, Canada, Korea, and Mexico, especially in the P1 period (2013-15), the TXF sample has a higher coverage for guarantees, covers a higher number of countries, and a higher degree of tranche-level granularity. To ensure that our findings are robust to differences in coverage, we conduct a robustness check using OCI values of ECA commitments wherever OCI data exceed TXF (see "Supplementary Methods"). Using this approach, we re-

calculate all figures presented in this article, except Fig. 4 for which OCI does not offer the sufficient level of granularity. As mentioned in the Discussion section, the overall trends and conclusions identified in this study remain robust to these checks.

## Inflation adjustment and income group classification

TXF reports financial volumes, such as the financial commitments by ECAs, in current USD. For comparability between years, we converted all nominal values to $USD_{2020}$ using the United States Consumer Price Index published by the International Monetary Fund[8,91]. We use the World Bank classification of income groups for the Fiscal Year 2024, which classifies countries according to their gross national income per capita in high-income, upper-middle income, lower-middle income and low-income countries[92]. With the term 'lower-income countries', we refer to lower-middle and low income countries.

## Data availability

The aggregate data generated in this study are provided in the Source data file. The raw transaction data used in this paper are proprietary and can only be obtained directly from TXF Limited (https://www.txfnews.com/). Access can be obtained by market participants or by researchers after reaching a data transfer agreement with the data provider. All other data used in this paper are publicly available at https://doi.org/10.5281/zenodo.14261240[93]. Source data are provided with this paper.

## Code availability

All scripts required to reproduce the analysis and the figures in this paper are publicly available at https://doi.org/10.5281/zenodo.14261240[93].

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

## Acknowledgements

We thank the team at TXF Limited for their keen support, advice, and numerous discussions on crucial data issues and the German Ministry for Economic Affairs and Climate Action for support with the data collection. We further thank practitioners at the Exile Global 2024 conference, staff members of the French Treasury, and participants at the 45th IAEE International Conference for valuable feedback and comments. This work benefitted from the European Union's Horizon 2020 research and innovation program, European Research Council (ERC), under grant agreement no. 948220, project GREENFIN (P.W.). It was conducted as part of the European Union's Horizon Europe research and innovation program project NEW PATHWAYS, and was supported by the Swiss State Secretariat for Education, Research and Innovation (SERI) under contract no. 24.00550 (B.S.). The opinions expressed and arguments employed herein do not necessarily reflect the official views of the European Commission or the Swiss Government.

## Author contributions

P.C. collected the data and conceived the study. P.C., P.W., I.S and B.S. developed the methodology. P.C. and P.W. re-classified the deals, and performed the data analysis, visualization, and triangulation. P.C. wrote the manuscript. P.W., I.S and B.S. reviewed and edited the manuscript.

## Funding

## Competing interests

P.C. and I.S. work with Perspectives Climate Research gGmbH, where they have previously conducted research on ECAs funded by Both ENDS, the European Climate Foundation, Oxfam America, and the Nordic Council of Ministers. These organizations have not been involved in the present study, which was led by the lead author's primary affiliation (HEC Lausanne). P.W. and B.S. do not declare any conflict of interest.
