## [Transparent Peer Review file · Nature Communications]

Quantifying the shift of public export finance from fossil fuels to renewable energy

Corresponding Author: Dr Paul Waidelech

Version 0:

Reviewer comments:

Reviewer #1

(Remarks to the Author)

The manuscript entitled "Quantifying the shift of public export finance from fossil fuels to renewable energy" addresses an important issue in today's economic and environmental landscape. The authors contribute to knowledge and practice with new and relevant results.

The introduction highlights the importance of tracking how export credit agencies (ECAs) are shifting towards financing renewable energy projects in response to global climate change initiatives. The methodology - explained in the annex - provides a clear overview of the approach used to analyse the shift. This is appropriate given the complexity of the financial data involved, but a brief discussion in the main text would help readers less familiar with these techniques.

While the manuscript provides insightful analysis and discussion of aggregate trends and the specific characteristics of deal financing across different technologies, it lacks a concise literature review. Including at least a brief discussion of recent studies would position the paper within the existing body of research, thereby enhancing its academic rigour and contextual relevance. In addition, the paper would benefit from a clearer presentation of the differences and implications of ECA activities between E3F members and non-members. This would help non-expert readers to fully grasp the nuances.

The discussion section skilfully outlines the main findings, but only briefly touches on limitations and potential avenues for further research. Expanding this section would provide a more balanced view of the scope and applicability of the research, particularly in suggesting how future studies could fill the gaps left by the limitations of the available data or methodology.

Overall, the paper is commendable for its contribution to understanding financial transitions in energy policy, but would have greater impact with these improvements. The recommendation is to accept with minor revisions.

(Remarks on code availability)

Reviewer #2

(Remarks to the Author)

General Comments:

The manuscript provides a thorough analysis of the shift in public export finance from fossil fuels to renewable energy sources over the period from 2013 to 2023. It utilizes data from export credit agencies (ECAs) across 32 countries, offering insights into technology trends, financing characteristics, and geographic implications. The study is well-structured, methodologically robust, and fills a significant gap in research on the role of ECAs in energy transitions.

Major Strengths:

The manuscript leverages a unique dataset of 911 energy deals, providing a robust foundation for a comprehensive analysis of export credit agencies (ECAs) and their financing activities. This dataset is particularly valuable due to the general challenges associated with accessing detailed transactional data in this sector. By examining underlying factors such as technological differences and policy impacts over time, the paper offers profound insights into the complexities of financial flows within the energy sector. Furthermore, the findings contribute significantly to ongoing policy debates about the role of

ECAs in the global energy transition, proposing evidence-based recommendations for re-aligning ECA mandates to support sustainable energy financing more effectively.

Major Concerns:

The manuscript effectively addresses the transition from fossil fuels to renewable energy within OECD countries, yet it lacks a comprehensive examination of developing nations. These countries often benefit from ECA funding but encounter unique economic and infrastructural challenges that require deeper analysis to fully understand their specific contexts and needs. Moreover, the study's robust data is somewhat compromised by limitations in coverage and potential biases, notably due to under-reporting from key countries such as China, Canada, and Mexico. Addressing these discrepancies is essential to enhance the generalizability of the findings. Additionally, the paper could achieve a more balanced viewpoint by incorporating perspectives from stakeholders who are skeptical about reducing fossil fuel financing. This should include an exploration of the economic repercussions for countries that depend heavily on fossil fuel exports, providing a more rounded discussion of the global energy transition's broader impacts.

Recommendations for Improvement:

To enhance the value and applicability of the study on export credit agencies (ECAs) and their role in the energy transition, it is crucial to broaden the geographic scope to include non-OECD countries. This expansion would provide a more balanced global perspective, shedding light on different economic contexts and the varied impacts of ECA strategies across a wider range of nations. Furthermore, improving data transparency is essential; this could be achieved through more extensive collaboration with international bodies and by securing comprehensive reporting commitments from ECAs. Such improvements in data handling would ensure a richer and more accurate analysis. Additionally, incorporating a broader spectrum of stakeholder perspectives, especially from those in developing countries and industries affected by shifts in ECA policies, would greatly enrich the study. Including these diverse viewpoints would offer deeper insights into the complexities of the global shift towards renewable energy, making the study's conclusions more robust and relevant.

(Remarks on code availability)

Version 1:

Reviewer comments:

Reviewer #2

(Remarks to the Author)

I have reviewed the recent revisions and accept the changes made by the authors. They have addressed the comments and suggestions satisfactorily, and the revisions have enhanced the clarity and quality of the manuscript. I have no further comments and approve the revised submission for consideration in its current form.

(Remarks on code availability)

**Authors' Response to Reviewers on
"Quantifying the shift of public export finance from fossil fuels to renewable
energy"
Manuscript Ref. No.: NCOMMS-24-17487**

We would like to thank the two anonymous reviewers for their valuable and constructive feedback. In response to the specific comments and suggestions for improvement, we have carefully revised our manuscript. While we provide detailed responses below, the major modifications to the manuscript can be summarized as follows:

- We sharpened the description of our methodology, enhanced the analysis of uncertainties involved by expanding the data triangulation in the Supplementary Information (SI), and discussed in greater detail the role of developing countries and of affected industries.
- As a result of the updated data triangulation, we excluded the Canadian ECA from the analysis (n=4 deals; its large domestic loan portfolio is not captured by TXF) and explain in detail why. For context, we report Canadian ECA commitments from a separate data source in the SI.
- We conducted a wide range of checks that led to improvements in the display and precision of our figures. This includes the amendment of a mistake in displaying the annual commitments for RETs in Fig. 2.
- By updating our data since the submission of our manuscript, our coverage increased by an additional n=14 deals (new sample size: n=921) in the area of green hydrogen and RET that have been closed in 2023.

Noting that none of the identified trends and conclusions of our study change in this revised manuscript, we present our responses to reviewers' comments below. We display the reviewer's comments in *italicized* font and our responses in blue. Where sensible, we highlight text additions to the initial manuscript in blue underlined.

Reviewer #1 (Remarks to the Author):

The manuscript entitled "Quantifying the shift of public export finance from fossil fuels to renewable energy" addresses an important issue in today's economic and environmental landscape. The authors contribute to knowledge and practice with new and relevant results.

The introduction highlights the importance of tracking how export credit agencies (ECAs) are shifting towards financing renewable energy projects in response to global climate change initiatives. The methodology - explained in the annex - provides a clear overview of the approach used to analyse the shift. This is appropriate given the complexity of the financial data involved, but a brief discussion in the main text would help readers less familiar with these techniques.

We agree with the reviewer that a brief discussion of our methodology in the main text makes the manuscript more accessible without referring to the Appendix. Therefore, we adopted the following changes in the Introduction:

Lines 41-43:

"ECAs support national exporters by issuing state-backed guarantees, thereby de-risking deals in overseas markets and mitigating repayment risks for private financial organizations. Outside of Europe, many ECAs also provide direct lending to deal participants.^{1,2}"

Lines 55-74:

“With this paper, we seek to fill in these gaps, drawing on commercial transaction data in export finance provided by TXF Limited, a private intelligence provider, that was previously unavailable for research. Using this novel data source, we identify a total of 921 energy deals between 2013 and 2023 in which ECAs from 31 OECD and non-OECD countries acted as either guarantor or direct lender. We exclude Canadian ECA energy finance from our analysis because it is substantially underreported in the TXF data and because the Canadian ECA acts more like a state investment bank, with most ECA finance flowing to projects in Canada rather than abroad.³ For more details on our data source and coverage, see **Methods** and the **Supplementary Information**. Based on transaction-level information, deal descriptions and additional desk research, we classify transactions into sectors and value chain stages following the approach by the Export Finance for Future (E3F) initiative, a climate club of ten European aimed to align public export finance with the objectives of the Paris Agreement.⁴ By analyzing financial commitments by ECAs across our global sample of energy-related transactions, we contribute to the extant literature in three ways. First, our paper is the first to provide an academic quantification of ECA energy finance flows with a global scope that covers long-term trends over the past decade. Second, we examine deal- and tranche-level information that allows us to determine the financing patterns of ECA countries by instrument, type of technology, and deal structures, among other characteristics. Third, we show the geographic implications of shifting energy portfolios, particularly for developing countries, and discuss resulting equity issues.”

Overall, we believe these additions provide more clarity about our methodology, while more detailed information is available in the Methods and the SI.

While the manuscript provides insightful analysis and discussion of aggregate trends and the specific characteristics of deal financing across different technologies, it lacks a concise literature review. Including at least a brief discussion of recent studies would position the paper within the existing body of research, thereby enhancing its academic rigour and contextual relevance.

To better position the paper within the existing body of research, we have now added a brief literature review and placed it in the SI (**Supplementary Table 4**), given that Nature Communications’ Article format does not feature a dedicated literature review section in the Main manuscript (except for the Introduction). Our new literature table complements the review of empirical papers and data sources that we had already provided in our original submission (**Supplementary Table 3**); for each previous paper it also describes the complementarity to the present manuscript. We refer the reader to both tables in the main text (lines 38-40): “Until today, a systematic, quantitative analysis of the relevance and characteristics of ECA energy financing is missing (see review of extant literature and data sources in **Supplementary Table 3 and 4**).”

Specifically, we identified the following twelve peer-reviewed articles on ECAs at the energy-climate nexus (for convenience, we copy **Supplementary Table 4** here with downsized font):

Article	Type of article	Data used	Most relevant findings	Complementarity of the present contribution
Klasen et al. (2024) ⁵ : Navigating geopolitical and trade megatrends: Public export finance in a world of change	Empirical and conceptual	Interviews, official documents and policy analysis	ECAs are adapting to rising geopolitical tensions, globalization, and climate change by shifting from traditional roles to proactive trade facilitators. ECAs increasingly align with national industrial policies, particularly in strategic sectors like climate-related technologies. Introduction of new financial products (e.g., untied loans, working capital guarantees) to meet evolving exporter needs.	This paper discusses broader shifts in the role of ECAs, which our paper supports with quantitative data and a specific focus on the energy sector.

Peterson and Downie (2024) ⁶ : The international political economy of export credit agencies and the energy transition	Commentary	Case studies of lending by the United Kingdom Export Finance Department (UKEF) for fossil fuel and renewable energy projects as well as aggregate data from Oil Change International.	ECAs have historically supported carbon-intensive sectors, but they also hold the potential to drive the clean energy transition. ECAs have been largely overlooked in the literature of International Political Economy (IPE), despite their substantial influence on global energy finance. Research should focus on how global and national climate governance efforts incorporate or fail to incorporate ECAs, and the effectiveness of such efforts.	This commentary highlights export finance in the energy sector as a major research gap in IPE scholarship, which we fill empirically with the present contribution.
Klasen and Vassard (2023) ⁷ : The new OECD arrangement on export credits: Breakthrough or bad compromise?	Commentary	Literature review and policy analysis	This analysis highlights the mixed outcomes of the new OECD Arrangement on export credits, recognizing it as a significant step forward while also pointing out areas where further reform and clarity are needed.	Our paper provides nuanced evidence of where the "green" shift within export finance in OECD countries and beyond is occurring and how.
Klasen et al. (2022) ⁸ : Export finance and the green transition	Empirical	Surveys from 20 EXIMs and ECAs Secondary data from publicly available sources, such as Climate Policy Initiative (CPI) EXIM and ECA annual reports for the period 2018–2020.	EXIMs and ECAs strongly support climate-action-related transactions with financing, guarantees, and insurance amounting to EUR 6.7–8.4 billion in 2020, surpassing previous CPI estimates. EXIMs and ECAs need to increase their climate financing 6.8 times, reaching EUR 45.3–57.4 billion by 2030, to meet the climate finance volumes required.	This paper provides context to our paper regarding the importance of ECAs for the energy transition and the need to scale up climate finance. Our paper provides comprehensive empirical data documenting where ECAs stand with regard to climate finance contributions.
Jansen (2022) ⁹ : The role of the OECD export credit arrangement	Empirical and conceptual	Aggregate historical data on export credits supported under the OECD Arrangement from 2007–2021 (OECD database) Reports and studies on the financing activities of the Chinese Development Bank and Chinese Export-Import Bank Analysis of recent policy developments and negotiations related to the green energy transition.	The OECD Arrangement on Officially Supported Export Credits plays a critical role in maintaining trade flows and supporting international trade, especially during times of crisis. The Arrangement has supported USD 717 billion in export credits from 2007 to 2021, with energy-related projects being a significant portion. There has been a shift from supporting fossil fuel projects to renewable energy projects, with renewable energy support increasing significantly since 2018. In 2020 and 2021, support for renewable energy projects was more than five times that of nonrenewable projects.	While this paper provides some high-level aggregate data on export finance in the energy sector based on the OECD database (and an aggregate on Chinese overseas energy financing), our paper complements it with a more granular and previously unpublished ECA finance data disaggregated by country, energy type, steps of the value chain, financial instrument, etc.
Lundquist (2022) ¹⁰ : Export credit agencies delivering finance for the green transition in times of crisis.	Empirical and conceptual	Case studies and experiences of EKF (Denmark's Export Credit Agency) Statistics on EKF's financing of renewable energy projects, especially wind energy Policy and regulatory framework analysis.	ECAs like EKF play a crucial role in financing the green transition by providing risk capital and long-term funding. ECAs need to continuously adapt their business models, products, and competencies to meet the evolving demands of the green transition. Continuous updating and recalibration of government policies and regulatory frameworks are necessary to incentivize green solutions and support the green transition.	This paper provides context to our paper regarding the importance of ECAs for the energy transition but, unlike our paper, covers only a single ECA.
Manych et al. (2023) ¹¹ : Pushed to finance? Assessing technology export as a motivator for coal finance abroad	Empirical and conceptual	Global Coal Plant Tracker World Electric Power Plants Data Base Global Coal Project Finance Tracker Semi-structured interviews.	Both public and private financial institutions continued to fund coal plants even after the Paris Agreement, although there is a noted downward trend in overall financial commitments. Primary Financing Countries: China, Japan, and South Korea are the dominant sources of cross-border debt financing for coal plants, particularly through public banks. Although countries like China, South Korea, and Japan have pledged to cease financing coal plants abroad, the implementation and adherence to these commitments remain uncertain.	This paper discusses the role of many different (private and public) actors involved in overseas investment in coal-fired power plants, one of which is ECA. Our paper expands this analysis by providing more granular data on coal deals and by also covering oil and gas, as well as renewable energy technologies and grid-related infrastructure.
Michie (2022) ¹² : The role of the global financial system in financing the transition to net zero	Commentary (practitioner)	Analysis of policy documents and case studies	While some ECAs have committed to net-zero targets by 2050, broader participation is needed. ECAs can learn from private sector initiatives like GFANZ to develop and implement effective net-zero strategies. ECAs should create comprehensive net-zero transition plans, similar to the private sector's adoption of climate-related financial disclosures. This would involve setting clear decarbonization strategies and targets.	The commentary discusses the policy reforms necessary to align ECAs with the Paris Agreement. Our paper provides quantitative empirical evidence to support it.
Hopewell (2021) ¹³ : Negotiating in the Dragon's Shadow: Export Credit for Coal Plants	Conceptual (book chapter)	Policy analysis	The absence of China, a major player in export credit for coal power plants, significantly undermines efforts to create effective global trade rules. This reflects broader difficulties in global governance when major economies are not part of key agreements. The reluctance of OECD countries to agree to restrictive measures without China's involvement limited the scope and ambition of the resulting agreement.	This book chapter highlights the importance of China's involvement in international policy processes around ECAs, for which our article provides empirical justification.
Liao (2021) ¹⁴ : The Club-based Climate Regime and OECD Negotiations on Restricting Coal-fired Power Export Finance	Empirical and conceptual	Primary data from interviews with people involved in the negotiations Secondary data from various sources including OECD reports and documents	The US led the negotiations to restrict export finance for coal-fired power projects within the OECD. Japan and South Korea resisted the US's push for stricter rules due to competitive disadvantages against Chinese ECAs not bound by OECD regulations. A 'club-based' approach to climate governance proved ineffective without the inclusion of powerful non-OECD countries like China.	This paper contextualizes the complexity of global climate governance and the need for inclusive and cooperative approaches that engage all major players, particularly China. Our paper builds on it by providing a quantitative analysis of the "green shift" in major OECD and non-OECD countries.
Hopewell (2019) ¹⁵ : How Rising Powers Create Governance Gaps: The Case of Export Credit and the Environment	Empirical	Interviews, policy analysis, case study	The rise of non-OECD countries as major exporters and financiers highlights the limitations of the current global governance framework, which was designed primarily for OECD countries. There is a pressing need to expand and adapt global governance mechanisms to include emerging powers and ensure that they adhere to environmental standards in their export credit activities.	This paper contextualizes the complexity of global climate governance and the need for inclusive and cooperative approaches that engage all major players, including non-OECD. Our paper builds on it by providing a quantitative analysis of the "green shift" in major OECD and non-OECD countries.
Wright (2011) ¹⁶ : Export Credit Agencies and Global Energy: Promoting National Exports in a Changing World	Conceptual	Aggregate OECD data on export credit agencies (ECAs) and their financing activities in the energy sector. Analysis of ECA policies.	ECAs significantly impact energy policy goals, including expanding energy supply in developing countries and influencing the carbon intensity of energy development. The growing influence of ECAs from non-OECD countries challenges the effectiveness and legitimacy of OECD governance arrangements.	This paper provides context to our paper regarding the importance of ECAs for the energy transition and discusses the limitations of OECD-based governance.

		Case studies and examples of ECA-supported projects in various countries.	Tensions exist between the national economic objectives of ECAs and their global environmental and social impacts.	
--	--	---	--	--

In addition, the paper would benefit from a clearer presentation of the differences and implications of ECA activities between E3F members and non-members. This would help non-expert readers to fully grasp the nuances.

We agree with the reviewer that the distinction between E3F and non-E3F countries is important to our analysis but could have been motivated more clearly in our initial manuscript. Therefore, we added the following paragraph to explain the nuances:

Lines 149-161:

“Launched in 2021, the E3F coalition consists of 10 European countries that pledged to align their export financing portfolios with the Paris Agreement.^{17,4} Specifically, this includes to “end new direct public support for the international unabated fossil fuel energy sector, except in limited and clearly defined circumstances in line with the 1.5° C scenario, by the end of 2022.¹⁸ In addition, E3F members now seek to increase and report on their ECA climate finance contributions, such as the new collective quantified goal of climate finance negotiated at COP29 in Azerbaijan.¹⁹ As a result, the E3F coalition is the only group of ECA countries that reports consistently on their energy finance and has a clear focus on policymaking in export finance. By contrast, some ECAs and their guardian authorities congregate in other climate initiatives in the sector¹², such as the Net-Zero Export Credit Agencies Alliance²⁰ or the Clean Energy Transition Partnership, an initiative following the Glasgow Statement at COP26²¹. We did not choose these initiatives as grouping categories since the former is more practitioner-oriented, and the latter a broader public finance initiative including non-ECA organizations.”

In addition, we added a broader discussion of E3F’s potential pioneering role in advancing international climate cooperation in the export finance sector (lines 284-292):

“Given the recent come-back of Chinese coal financing²², policymakers should hence consider relaunching the Working Group for climate cooperation. In this context, the E3F coalition could play a pioneering role in demonstrating which policies can incite exporters to pivot into sustainable markets, for instance, by aligning ECAs’ mandates with the Paris Agreement and offering favorable terms and conditions for Paris-aligned projects. The finding that, since 2022, joint annual energy sector commitments by E3F countries are almost on par with non-E3F countries (USD₂₀₂₀ 13.3 billion versus USD₂₀₂₀ 14.9 billion) underlines the growing importance of this coalition.”

The discussion section skillfully outlines the main findings, but only briefly touches on limitations and potential avenues for further research. Expanding this section would provide a more balanced view of the scope and applicability of the research, particularly in suggesting how future studies could fill the gaps left by the limitations of the available data or methodology.

We thank the reviewer for this comment and agree that more nuances should be provided regarding limitations and future research avenues. To expand the discussion section accordingly, we clarified the following limitations and discussed how to potentially address them in future research:

Limitation 1: TXF has partially incomplete data coverage, especially concerning direct lending to fossil fuel projects by the Canadian ECA, but also by ECAs from China, Korea, Japan and Mexico. This was evident from the data triangulation presented in

Supplementary Fig. 1-3 in the initial submission manuscript. Using an updated dataset by the NGO Oil Change International (OCI), we now conducted a series of further analyses for Figures presented in the main manuscript, except for **Fig. 4**, which uses deal- and tranche-level information that OCI does not report. Against this background, we direct readers interested in the highest deal coverage (over transactional detail) to the SI.

In this context, we modified lines 341-346, highlighting that our results display global coverage excluding Canada:

“While our sample exhibits an unprecedentedly high degree of coverage regarding countries and guarantee volumes, there are important limitations to our analysis that leave gaps for future research. Most pertinently, our results exclude Export Development Canada, the Canadian ECA, and likely under-report some countries that do not completely report deals to TXF, e.g., China and Mexico (see **Supplementary Fig. 2a and 2b**).”

We discuss implications for our results and policy in the newly added paragraph in lines 346-358:

“As a corollary, our results present a lower bound of ECA financing, especially concerning direct ECA lending to fossil fuel projects. To address this limitation, we conduct a robustness check in which we rely on data gathered by the nongovernmental organization Oil Change International wherever TXF deals appear missing or incompletely reported (see **Methods and Supplementary Fig. 1-7**). While the overall trends and conclusions identified in this study remain consistent, the robustness check suggests that the rise of RET share in overall ECA energy finance may be somewhat less pronounced and the magnitude of total ECA energy commitments significantly higher, particularly between 2013 and 2015. These remaining uncertainties underpin the need for public policy efforts to improve data transparency, especially concerning the transaction databases hosted by the OECD and the Berne Union, the leading global association for the export credit and investment insurance industry, both of which - much unlike other public finance institutions - remain inaccessible to the public.”

Limitation 2: We only analyze public ECAs, i.e., government-owned/-sponsored finance agencies that can directly be influenced by public policy, and only study ECA commitments for the energy sector and not other climate-relevant sectors. We discuss this limitation and call on future research in the newly added lines 359-364:

“First, the scope of our study is restricted to public ECAs and the energy sector, while public ECAs represent only half of the Berne Union’s member organizations.²³ Future research could therefore explore the role of private export credit and investment insurance agencies, as well as the role of ECAs in de-risking infrastructure in non-energy but climate-relevant sectors, such as mining, industry, and transport.”

Limitation 3: Our study provides descriptive evidence only, i.e., based on our observational data we do not identify causal or second-order effects of ECA finance. We outline avenues for future research to fill in these gaps and modify lines 364-368:

“Second, our study provides descriptive evidence on ECA financing patterns following policy events, such as the OECD-wide ban of ECA support for coal-fired electricity generation, but it does not identify the causal impacts of policy changes. Here, future research could use quasi-experimental designs to isolate the effects of specific policy measures from broader trends.”

Furthermore, we add lines 368-373:

“Third, we do not examine economy-wide second-order effects of shifting ECA portfolios towards RETs, for instance, on employment or public budgets. To fill in these gaps, case studies using qualitative methods and modeling studies could help to better understand context-dependent second-order effects and risks and opportunities for affected industries, especially in economies that depend on exports of fossil fuels or equipment used in oil and gas value chains.”

Limitation 4: Our study focuses on ECA-supported deals only and we do not conduct a systematic comparison with (similar) non-ECA deals. We briefly outlined avenues for future research to address this gap in the newly added lines 373-376:

“Fourth, our study only examines energy deals with ECA involvement, whereas future research could systematically compare deals with and without ECA involvement to identify drivers of ECA involvement (data availability currently limits our analysis in that regard).²⁴”

Limitation 5: Finally, we highlight the backward-looking approach of our analysis, which considers closed deals from past years, and sketch an avenue for future research to address this gap in lines 376-385:

“Finally, our study adopts a backward-looking approach by analyzing closed deals. Regarding the financing needs of low-carbon transitions in developing countries in particular, scholars could complement our study using forward-looking methods and examine the potential role of ECAs in meeting these needs. One way could be using computational models that quantify energy system investments under different scenarios, and consider ECA effects through a reduction of project-specific cost of capital (which have a markable effect on projected investment volumes especially in developing countries). To calibrate such forward-looking analyses, more empirical work is needed concerning the precise impact of ECA involvement on financing conditions for both fossil fuel and renewable energy projects.²⁵⁻²⁷”

Overall, the paper is commendable for its contribution to understanding financial transitions in energy policy, but would have greater impact with these improvements. The recommendation is to accept with minor revisions.

We thank the reviewer for their positive assessment of our work and their valuable suggestions for improvements.

Reviewer #2 (Remarks to the Author):

General Comments:

The manuscript provides a thorough analysis of the shift in public export finance from fossil fuels to renewable energy sources over the period from 2013 to 2023. It utilizes data from export credit agencies (ECAs) across 32 countries, offering insights into technology trends, financing characteristics, and geographic implications. The study is well-structured, methodologically robust, and fills a significant gap in research on the role of ECAs in energy transitions.

Major Strengths:

The manuscript leverages a unique dataset of 911 energy deals, providing a robust foundation for a comprehensive analysis of export credit agencies (ECAs) and their financing activities. This dataset is particularly valuable due to the general challenges associated with accessing detailed transactional data in this sector. By examining underlying factors such as technological differences and policy impacts over time, the paper offers profound insights into the complexities of financial flows within the energy sector. Furthermore, the findings contribute significantly to ongoing policy debates about the role of ECAs in the global energy transition, proposing evidence-based recommendations for re-aligning ECA mandates to support sustainable energy financing more effectively.

Major Concerns:

The manuscript effectively addresses the transition from fossil fuels to renewable energy within OECD countries, yet it lacks a comprehensive examination of developing nations. These countries often benefit from ECA funding but encounter unique economic and infrastructural challenges that require deeper analysis to fully understand their specific contexts and needs. Moreover, the study's robust data is somewhat compromised by limitations in coverage and potential biases, notably due to under-reporting from key countries such as China, Canada, and Mexico. Addressing these discrepancies is essential to enhance the generalizability of the findings.

We agree with the reviewer that data coverage and underreporting represent challenges to establishing conclusive evidence for ECA energy finance flows. To address this point, our initial manuscript featured a supplementary analysis, in which we imputed values from Oil Change International (OCI), a non-governmental organization, to 'fill up' volumes where deals appear under- or un-reported in our original TXF sample. As we show in our **Supplementary Fig. 2a-b**, TXF covers many more countries (n=31) and has a particularly good coverage for credit guarantees. Meanwhile, OCI reports significantly higher financing volumes for direct lending to fossil fuel projects by ECAs from Canada, China, Korea, Japan, and Mexico. In the initial submission, we had already included an analysis where we filled up missing TXF values using OCI data at the year level (retaining the highest values in each year from *either* TXF or OCI data).

Considering the reviewer's concerns, we have now refined this analysis by retaining the higher value from either TXF or OCI for each pairing of technology, ECA country and year (e.g., all German ECA financing for natural gas in 2013), and an additional fourth dimension that is either the financial instrument type (direct lending or guarantee), the recipient country where the financed projects are located, or the value chain stage (e.g., upstream or downstream). Using this approach, we replicate all our main figures using joined TXF and OCI data, except for **Fig. 4**, which requires deal- and tranche-level information that OCI does not report, such as loan tenors or borrower types. In doing so, we now use an updated dataset that was obtained in July 2024 in direct liaison with data experts from OCI and that, unlike the version used in our initial submission, also covers transactions that closed in 2022. We provide more information on the dataset and method used to join it with TXF data in the **Note on Supplementary Fig. 1-7** in the SI.

The results of the combined datasets are presented in the updated and newly added **Supplementary Fig. 1-7**. While the general conclusions concerning the ongoing transition towards RETs of our analysis do not change, the triangulated figures underline that ECAs in several countries have supported fossil fuel projects with significantly higher volumes than those reported to TXF (see **Supplementary Fig. 2a-b**). Given the particularly stark discrepancy between TXF and OCI for Canadian commitments and the high domestic financing levels of the Canadian ECA Export Development Canada (untypical for ECAs), we decided to exclude this institution from the scope of our manuscript and to make very transparent that the analysis excludes the Canadian ECA (which we mention in our Abstract, Introduction, Results and Methods section). Effectively, this excludes only $n = 4$ deals from the main analysis in which Export Development Canada was the sole ECA deal participant and a cumulative total of USD₂₀₂₀ 2.8 billion. While coverage issues from Canada persist over time, in general, the differences in coverage of other countries between TXF and OCI decrease substantially over time (see **Supplementary Fig. 1**).

In light of these robustness checks, we added qualifications to conclusions drawn in the main manuscript, where appropriate (see line numbers in Table below). In summary, this supplementary analysis provides confidence in the Main manuscript's results while highlighting that the reported ECA energy finance flows for fossil fuel may represent lower-bound estimates for some countries.

Conclusion main manuscript	Re-evaluation and implications	References
"[...] the share of ECA energy commitments to RETs has markedly increased from 9% and 5% in 2013 to a two-year average of 42% and 39% for guarantees and direct lending in 2022-2023, respectively" (see lines 103-105, Main) (Aggregate trends)	Re-evaluation Considering the joined data frames of TXF and OCI in the last year of full reporting (2022), we find a RET share global ECA energy commitments rises to 35% (guarantees) and 25% (direct lending). Implications Since we cannot triangulate TXF data for 2023 (last complete reporting year in the OCI data is 2022), we undertook the following precautions in the main manuscript:  1) We report the average of both years in the main manuscript that we deem most robust. 2) Unlike in the earlier version of this manuscript, we refrain from presenting exact RET shares in the abstract. We discuss implications of this discrepancy in the Discussion of the main article, and that we await better transparency and data updates to reduce the uncertainty.	Supplementary Fig. 3, Source Data
"[...] between 2013 and 2023, total ECA commitments to the energy sector varied between USD₂₀₂₀ 22 and 43 billion per year" (see lines 91-92, Main) (Aggregate trends)	Re-evaluation Considering the joined data frames of TXF and OCI, we find that especially in P1 (2013-2015) total financing commitments would be considerably higher (see Supplementary Fig. 3), especially when including Canada (see Supplementary Fig. 7). For an average year, TXF omits annual energy commitments of USD₂₀₂₀ 12 billion from Export Development Canada (EDC). We estimate that about 90% of EDC's commitments are disbursed domestically³ which is a plausible explanation why TXF does not capture EDC's high commitments. Implications We added the following sentence to the main manuscript:	Supplementary Fig. 1 and 3, Source Data

	“Especially during P1 (2013-2015), TXF very likely under-reports real total ECA energy commitments as we show in Supplementary Fig. 1, 3 and 7” (lines 93-94).	
“The subsequent, massive drop in gas financing post-Glasgow (P4) is potentially due to post-pandemic risk aversion for long-term commitments in supporting larger-scale upstream or downstream projects²⁸. An additional explanation is that large untied state guarantees to secure foreign gas supplies (e.g., in the case of Germany²⁹) are not included [...]” (see line 124-128, Main) “The sharp fluctuations in ECA support for fossil fuels over the recent years contrast with the steadily growing RET commitments” (see line 130-131, Main) (Trends along the fossil fuel value chain and by technologies)	Re-evaluation Considering the joined data frames of TXF and OCI, we find less pronounced changes in gas upstream and RET financing. This specifically includes:  1) The differences between P3 (2020 and 2021) and previous periods is less pronounced. The significant subsequent drop in P4 (2022) remains valid (Supplementary Fig. 4). 2) High volumes of Chinese RET financing are omitted in TXF data, especially in early years of the observation period. Implications No major implications for the identified relative trends (since only absolute commitment volumes may be understated in the Main manuscript). Note also that Supplementary Fig. 4 reports 2022 data only, while 2023 has been the year with the highest RET commitments to date according to TXF data.	Supplementary Fig. 2a and 2b, Supplementary Fig. 4, Source Data
“[...] compared to fossil fuel finance, ECA support for RET projects is much more likely to remain in the same region” (see line 241-242, Main) (Geographic implications)	Re-evaluation Considering the joined data frames of TXF and OCI, this finding holds for all ECA countries, except for Chinese RET commitments: OCI data shows that Chinese RET commitments almost exclusively go abroad. Implications Since this is a country-specific implication, we did not carry out any respective changes to the Main manuscript in this regard.	Supplementary Fig. 6, Source Data

Importantly, we exclusively use TXF data in the Main manuscript due to the availability of deal- and tranche-level characteristics, which is a pre-requisite for the analysis presented in **Fig. 2** and **Fig. 4** and the much higher coverage of countries (31 countries compared to only 16 countries covered by OCI). In addition, combining different databases can help to assess potential under-reporting but comes with substantial risks of inconsistencies and potential double counting. This is why the results presented in the main text remain based only on TXF data. We hope that the additional supplementary material and the text changes in the main text help to alleviate the reviewer’s concerns regarding the generalizability of our findings. The revised Discussion section further emphasizes the need for better data transparency and consistent reporting across countries to enhance fruitful research avenues in the future.

Finally, as we discuss at more length in our reply to another reviewer comment below, the revised manuscript includes a nuanced discussion of the role of developing countries. Specifically, we highlight and contextualize the observation that (low-income) developing

countries are rarely host countries of ECA-supported energy projects in the first place (lines 301-306):

“More generally, the observation that ECAs rarely support deals in low-income countries can be explained by economic considerations. Unlike development finance institutions whose mission is to support socio-economic development, ECAs traditionally help national exporters to realize deals primarily based on commercial profitability. This is scarcely the case in low-income countries that typically require highly concessional conditions, e.g., from multilateral entities at the World Bank Group.³⁰”

We further mention the notable exception of Mozambique and describe it (and lines 251-252; also see discussion below):

“The only significant exception in our sample is Mozambique, one of the world’s poorest countries, which accounts for nearly 22% of ECA energy commitments in P3.”

Together with the generally observed shift of ECA energy finance towards high-income countries, this warrants to recommend that (lines 306-308):

“[o]ne way to counterbalance the geographical trends identified in our study could be to expand ECA mandates by explicitly supporting the energy transition in developing countries.^{31,32}”

And we add that, given the heterogenous needs of developing countries to shape their clean energy future³³, (lines 337-339):

“[p]olicy-makers should therefore prioritize phase-out policies that sufficiently consider the project country context, thus ensuring a just transition.^{31,33,34}”

We further added important related elements to the Discussion, including diverse perspectives on reducing fossil fuel financing, the role of China and other non-OECD countries, and implications for fossil fuel-dependent countries (see our reply to other comments below).

Additionally, the paper could achieve a more balanced viewpoint by incorporating perspectives from stakeholders who are skeptical about reducing fossil fuel financing. This should include an exploration of the economic repercussions for countries that depend heavily on fossil fuel exports, providing a more rounded discussion of the global energy transition's broader impacts.

We thank the reviewer for this comment and agree that a broader discussion of stakeholder perspectives is beneficial. To this end, we have added an analysis of highly fossil fuel export-dependent countries (**Supplementary Note 2**). Using a composite measure of defining ‘fossil fuel export-dependent’, we identify these countries and calculate the share of ECA commitments received by them between 2013 and 2023. To better understand the associated risks of shifting ECA finance away from fossil fuels, we added the following paragraph to the Discussion (lines 309-339):

“ECA climate policies can have important adverse implications for traditional export industries. In line with evidence showing export technologies being a major ‘push factor’ of financing activities¹¹, ECA portfolios typically reflect the composition of national export industries.¹⁸ In most countries, this historically included goods and services related to the production, transport or use of fossil fuels, rather than RETs. However, instead of largely remaining demand-driven agencies, there are important calls for ECAs to become active agents in the pursuit of reconciling climate, industrial

and trade policy objectives.⁵ Indeed, some ECAs already support ventures into emerging green markets that require technical capabilities similar to those of traditional export industries.³⁵ In 2023, for instance, the German ECA provided cover to build the first large-scale green hydrogen and ammonia facility in Saudi Arabia³⁶ and the first large-scale green steel factory in Sweden.⁴⁰ Regarding employment in traditional energy export industries, input-output modeling suggests overall positive employment effects of investment in more labor-intensive RETs⁴¹ and specifically for shifting ECA support towards RETs for some countries^{42,43}. However, historical experiences of transitions away from coal highlight the economic risks, especially in the labor market.⁴⁴ To soften employment impacts, withdrawing ECA support for fossil fuel technologies could be combined with additional policies facilitating the transition of workers to cleaner exporter industries, e.g., retraining and compensation schemes.⁴⁵

Besides traditional export industries, ending ECA financing for fossil fuels can also affect host countries that are economically dependent on fossil fuel exports, such as the United Arab Emirates or Mozambique.⁴⁶ In our sample, a significant share (40%) of total ECA energy finance flows to fossil fuel-dependent host countries, mostly supporting fossil fuels rather than RETs or the grid sector (see **Supplementary Note 2**). Such countries may advocate for longer transition periods and continued eligibility for ECA support in the fossil fuel sector to attenuate the potentially adverse economic effects of fossil fuel phase-outs.^{33,46,47} This may include oil and gas workers whose skills are not directly transferrable to other sectors, e.g., in the highly specialized and labor-intensive parts of the upstream industry.⁴⁸ Yet, besides the climate and environmental risks of new fossil fuel infrastructure^{49,50}, continued eligibility may also increase risks of asset stranding.⁴⁷ Policy-makers should therefore prioritize phase-out policies that sufficiently consider the project country context, thus ensuring a just transition.^{31,33,34}

Recommendations for Improvement:

To enhance the value and applicability of the study on export credit agencies (ECAs) and their role in the energy transition, it is crucial to broaden the geographic scope to include non-OECD countries. This expansion would provide a more balanced global perspective, shedding light on different economic contexts and the varied impacts of ECA strategies across a wider range of nations.

We thank the reviewer for this important suggestion and agree that understanding the role of non-OECD countries is crucial here. Historically, early industrialized countries have first created ECAs to support their exporters abroad. Therefore, the majority of our sample are OECD countries (22 out of 31 countries), but our initial manuscript already included ECAs from nine non-OECD countries, listed in **Supplementary Table 1**:

1. China (Sinosure and Chexim)
2. India (India Eximbank and ECGC)
3. Indonesia (Indonesia Eximbank)
4. Malaysia (Mexim)
5. Russian Federation (Exiar and Roseximbank)
6. Saudi Arabia (Saudi Exim Bank)
7. South Africa (ECIC)
8. Thailand (Thai Eximbank)
9. United Arab Emirates (ADEX)

However, the reviewer's comment illustrates that our initial manuscript did not communicate our sample composition clearly enough. Therefore, we have clarified that our sample covers OECD and non-OECD countries alike in our Abstract (line 18) and in our Introduction (lines 58).

What our initial manuscript also did not do was analyze OECD and non-OECD countries separately to identify potential differences. Therefore, our revised manuscript now splits the non-E3F member group in **Fig. 3B** into OECD and non-OECD countries. We find that unlike for most OECD countries, there is no clear trend of 'portfolio greening' among non-OECD countries over the past decade. As overall volumes are dominated by China, we also include a new country-by-country breakdown of non-OECD and OECD countries in the SI (**Supplementary Table 6 and 7**, respectively). This overview, which uses the filled-up data combining TXF and OCI data, shows that 80% of total commitments between 2013 and 2023 originate from OECD countries. Furthermore, within the non-OECD group, 85% of total energy commitments come from China only.

We have incorporated these additional findings into our discussion of **Fig. 3** accordingly, highlighting the predominant role of China outside the OECD, but also, and increasingly, as a main counterweight to traditionally strong ECA countries from within the OECD, such as Korea and Japan (modified lines 276-286):

"Outside the OECD, we find that especially Chinese ECAs, Sinosure and the Export-Import Bank of China, dominate ECA energy financing, albeit with an increasing share of fossil fuels (Fig. 3B, also see Supplementary Fig. 2b and Supplementary Table 6). Chinese ECAs today represent a major counterweight to traditional ECAs, and fears of giving away market shares to China inhibit (climate) policy changes within the OECD, especially from regional competitors South Korea and Japan.^{14,51,52} In general, ECA policies remain uncoordinated between OECD and non-OECD countries, given disagreements about transparency issues which caused the suspension of the International Working Group on Export Credits, the only transpacific forum for public export finance policies previously led by the United States and China.⁵³ Given the recent come-back of Chinese coal financing²², policymakers should hence consider relaunching the Working Group for climate cooperation."

Considering the new results in the SI, we think that these changes provide a more balanced perspective on the geographic scope of ECA energy finance and we thank the reviewer for suggesting that we improve our manuscript in this regard.

Furthermore, improving data transparency is essential; this could be achieved through more extensive collaboration with international bodies and by securing comprehensive reporting commitments from ECAs. Such improvements in data handling would ensure a richer and more accurate analysis.

We agree with the reviewer that data transparency regarding ECAs' energy finance is key and increased transparency and collaboration by ECAs and internal organizations would be invaluable in improving accuracy. This is one of the reasons that motivated our research and we hope the present study can help highlight the need for it. There are two international bodies with data on the export and investment insurance industry. The first one, the OECD Export Credit Group, however, only keeps books for ECAs from OECD countries and is therefore insufficient for a global quantification of ECA energy finance flows, as conducted in our study. In addition, upon consultation with the OECD Export Credit Group, the level of granularity necessary for our analysis cannot be made available for our purposes given the confidentiality of transaction-level data. The second one, the Berne Union, the umbrella association for the global export credit and investment insurance industry, keeps a transaction database from 83 institutional members. However, Berne Union's reporting is too coarse to robustly identify energy-related transactions and classify them into specific sectors and technologies. We have attempted to gain access to non-public reporting data collected by the Berne Union, but this has also been unsuccessful since the confidentiality of the data involved would have placed too many restrictions on our analysis. Therefore, both avenues to improve data availability and transparency through international bodies proved unfeasible in our case.

These issues notwithstanding, our work is the first to analyze TXF data, the main commercial transaction data provider in the export finance community, a source that was previously unavailable for research. This represents a major data advancement and substantially exceeds the level and depth of analysis of previous studies. Therefore, we believe that our manuscript still holds great value for the academic community, policymakers and practitioners, particularly given the lack of a global analysis of ECAs' energy finance in the extant literature (see our review of empirical work and literature in **Supplementary Tables 3 and 4** and triangulation to evaluate coverage in **Supplementary Fig. 2a-b**).

In response to the restricted transparency in the export finance community (especially compared to the more transparent reporting practices by multilateral and other public development banks), we added a discussion of data limitations to our revised Discussion (lines 354-358). Specifically, we add that:

"These remaining uncertainties underpin the need for public policy efforts to improve data transparency, especially concerning the transaction databases hosted by the OECD and the Berne Union, the leading global association for the export credit and investment insurance industry, both of which - much unlike other public finance institutions - remain inaccessible to the public."

We sincerely hope that future policy efforts can address these important data gaps.

Additionally, incorporating a broader spectrum of stakeholder perspectives, especially from those in developing countries and industries affected by shifts in ECA policies, would greatly enrich the study. Including these diverse viewpoints would offer deeper insights into the complexities of the global shift towards renewable energy, making the study's conclusions more robust and relevant.

Acknowledging the fact that the observed energy transition has heterogeneous impacts on a variety of stakeholders and nations, we fully agree with the reviewer's observation to discuss a broader spectrum of repercussions. As laid out above, we have added potential adverse implications for traditional export industries and fossil fuel-dependent countries to our Discussion section. In addition, we have expanded the discussion of the role of developing countries. Specifically, we deepened the interpretation of **Fig. 5** which shows the distribution of ECA funds among recipient countries. One decisive observation is that low-income countries, except for Mozambique, have received almost no ECA support for developing any energy projects. Given the salience and controversy around the Mozambiquan case, we add a brief discussion of this deal (lines 251-260):

"The only significant exception in our sample is Mozambique, one of the world's poorest countries, which accounts for nearly 22% of ECA energy commitments in P3. Here, several ECAs supported the stepwise development of natural gas reserves estimated at up to 4,247 billion cubic meters that were discovered in 2010⁵⁴, about twelve times Europe's 2023 annual gas demand.⁵⁵ With total commitments of USD₂₀₂₀ of 13.8 billion for Mozambique LNG in 2020, this project alone represents about four times Mozambique's average foreign direct investment between 2018 and 2022.⁵⁶ However, escalating violence following insurgency attacks in the region led the project developers to suspend Mozambique LNG in 2021.⁵⁷ As of 2024, however, some industry observers estimate that, in 2025, the project may come back on track together with an even larger sister deal to develop adjacent fields.⁵⁸"

We further discuss the observed trends and patterns in the Discussion section (lines 293-308):

"Beyond trends and patterns in climate-relevant ECA activities, our results highlight strong discrepancies between developed and developing nations, with the former

accounting for most ECA commitments (see **Supplementary Tables 6 and 7** for a split between OECD and non-OECD country financing). Aside from China, non-OECD countries in our sample, such as Indonesia, Russia, or Saudi Arabia, established their ECAs only in the 1990ies or 2000s and show comparatively little but growing ECA activity.¹⁶ On the recipient side, we find that ECA portfolios markedly shift to high-income countries, as the energy transition is ongoing. This is particularly concerning for the clean energy financing needs of emerging and developing nations, e.g., on the African continent.^{33,59} More generally, the observation that ECAs rarely support deals in low-income countries can be explained by economic considerations. Unlike development finance institutions whose mission is to support socio-economic development, ECAs traditionally help national exporters to realize deals primarily based on commercial profitability. This is scarcely the case in low-income countries that typically require highly concessional conditions, e.g., from multilateral entities at the World Bank Group.³⁰ One way to counterbalance the geographical trends identified in our study could be to expand ECA mandates by explicitly supporting the energy transition in developing countries.^{31,32}

Finally, as part of the research process, the first author (P.C.) presented the study at the Exile Global 2024 conference in Athens (June 10th to 12th 2024), the major export, development and commodity finance conference worldwide. In this context, we discussed the risks and opportunities of shifting to RETs with a wide range of stakeholders, including commercial banks, ECAs, exporters, and governments. We considered these diverse perspectives when drafting this revised version of the manuscript.

Further changes to the manuscript:

As part of our revision, we identified and corrected several coding bugs without implications for the identified trends and conclusions and updated the underlying data where possible. As a result, we carried out the following eight changes independently of reviewer comments:

#	What	Why	Implications
1	Corrected volumes of Renewables financing volumes in Fig. 2 .	A coding bug led to wrong Renewables volumes in Fig. 2 of the initial submission.	Correcting the volumes required the following changes in the main manuscript: • Lines 131-133: RET commitments in P4 (2022-2023) are now for the first time higher than commitments for all types of fossil fuels.• Lines 140-143: Volumes of total power-related volumes displayed (instead of an average annual of USD₂₀₂₀ 10 billion, we now find USD₂₀₂₀ 12.9 billion). We provide a comparison of the initial Fig. 2 and the updated Fig. 2 underneath this table (see below). We also carefully reviewed all code using the four-eyes principle.
2	Increased sample size from 911 to 925 deals.	Sample and category updates by TXF.	Minor implications for the overall volumes, number of deals and tranches specified for each figure. Adds a new outlier data point to Fig. 4a (the USD₂₀₂₀ 8 billion NEOM Green Hydrogen Project in Saudi Arabia).
3	Additionally classified the value chain stage of thirteen deals.	Due to a coding bug, these deals were previously missing in Fig. 2 .	Barely visible changes in Fig. 2. Added commitments in USD₂₀₂₀ amount to: - 4 million for oil (Power generation)- 5 million for gas (Power generation)- 1 million for coal (Power generation)- 18 million for coal (Upstream). Also see comparison Fig. 2 below.
4	Corrected a typo in the 'Note' describing the	The P2 period was erroneously labelled as "2015-	No further implications.

	periods in Fig. 2 and 3 .	2019", instead of "2016-2019".	
5	Corrected a coding bug in counting and categorizing the number of tranches per borrower type (Fig. 4C) and financial instrument (Fig. 4D).	Some (rare) borrower categories were erroneously omitted from the borrower type/financial instrument counts in Fig. 4 .	Barely visible changes in Fig. 4C and 4D . See Source Data for updated results.
6	Correction of OCI volumes displayed in initial SI.	Throughout the revisions, we incorporated the latest version of the OCI data, which included updated variable descriptions. We thus noticed that we had excluded the category 'Mixed' instruments in our initial SI submission.	This category includes commitments that cannot directly be attributed to direct lending or guarantees (especially by Canada and South Korea) and is significant (USD₂₀₂₀ 100 billion). We amended this issue in the updated SI (see Note on methods for Supplementary Fig. 1-7) and included a separate category for such mixed/other instruments in Supplementary Fig. 3 and Supplementary Fig. 7. The main implication is higher volumes (fossil), especially in the year 2014 (Supplementary Fig. 3).
7	Updated inflation adjustment	The IMF published US Consumer Price Index data that covers the entire year of 2023, which we used to re-calculate the inflation adjustment.	Update of deflators used, which resulted in barely visible changes to the displayed figures. See Source Data and GitHub Repository for updated figures.
8	Corrected a typo about Japanese annual financing volumes	Shift in Japanese financing volumes before and after the pandemic was given as -15%, while the correct value is -50% (lines 167-171).	No further implications.

Comparison Fig. 2 (referred to in #1 in the above Table).

Fig. 2 (updated submission).

Fig. 2 (initial submission).

References:

1. Stephens, M. *The changing role of export credit agencies*. (International Monetary Fund, Washington, DC, 1999).
2. Shishlov, I., Weber, A.-K., Stepchuk, I., Darouich, L. & Michaelowa, A. External and internal climate change policies for export credit and insurance agencies. *CIS Work. Pap.* (available at: <https://www.zora.uzh.ch/id/eprint/186777/>) (2020).
3. Censkowsky, P., Shishlov, I. & Darouich, L. Paris alignment of export credit agencies: The case of Canada (Export Development Canada). (available at: <https://perspectives.cc/publication/paris-alignment-of-export-credit-agencies-canada-export-development-canada/>) (2022).
4. Export Finance for Future. E3F status report 2023. (available at: https://www.ekn.se/globalassets/dokument/hallbarhetsdokument/e3f_status_report_2023.pdf) (2023).
5. Klasen, A., Krummacker, S., Beck, J. & Pennington, J. Navigating geopolitical and trade megatrends: Public export finance in a world of change. *Glob. Policy* **forthcoming** (early view available at: <https://doi.org/10.1111/1758-5899.13417>) (2024).
6. Peterson, M. & Downie, C. The international political economy of export credit agencies and the energy transition. *Rev. Int. Polit. Econ.* **31**, 978–994 (2024).
7. Klasen, A. & Vassard, J. The new OECD arrangement on export credits: Breakthrough or bad compromise? *Glob. Policy* **14**, 958–961 (2023).
8. Klasen, A., Wanjiru, R., Henderson, J. & Phillips, J. Export finance and the green transition. *Glob. Policy* **13**, 710–720 (2022).
9. Jansen, M. Managing the green transition The role of the OECD export credit arrangement. *Glob. Policy* **13**, 554–556 (2022).
10. Lundquist, P. Export credit agencies delivering finance for the green transition in times of crisis. *Glob. Policy* **13**, 530–533 (2022).
11. Manych, N. *et al.* Pushed to finance? Assessing technology export as a motivator for coal finance abroad. *Environ. Res. Lett.* **18**, 084028 (2023).
12. Michie, A. The role of the global financial system in financing the transition to net zero. *Glob. Policy* **13–4**, 557–562 (2022).
13. Hopewell, K. Power transitions and global trade governance: The impact of a rising China on the export credit regime. *Regul. Gov.* **15**, 634–652 (2021).
14. Liao, J. The club-based climate regime and OECD negotiations on restricting coal-fired power export finance. *Glob. Policy* **12**, 40–50 (2021).
15. Hopewell, K. How rising powers create governance gaps: The case of export credit and the environment. *Glob. Environ. Polit.* **19**, 34–52 (2019).

16. Wright, C. Export credit agencies and global energy: Promoting national exports in a changing world. *Glob. Policy* **2**, 133–143 (2011).
17. French Treasury. Seven countries launch international coalition “Export Finance for Future” (E3F) to align export finance with climate objectives. (available at: <https://www.tresor.economie.gouv.fr/Articles/2021/04/14/seven-countries-launch-international-coalition-export-finance-for-future-e3f-to-align-export-finance-with-climate-objectives#:~:text=During%20a%20virtual%20meeting%20on,coalition%20to%20harness%20public%20export>) (2021).
18. Export Finance for Future. E3F transparency reporting 2022. (available at: <https://www.ekn.se/globalassets/dokument/hallbarhetsdokument/e3f-transparency-report-2022.pdf>) (2022).
19. Falduto, C., Noels, J. & Jachnik, R. The new collective quantified goal on climate finance: Options for reflecting the role of different sources, actors, and qualitative considerations. *OECD-IEA Clim. Change Expert Group Pap.* 2024/02. (available at: https://www.oecd-ilibrary.org/environment/the-new-collective-quantified-goal-on-climate-finance_7b28309b-en) (2024).
20. United Nations Environment Program's Finance Initiative. Net-zero export credit agencies alliance. (available at: <https://www.unepfi.org/climate-change/net-zero-export-credit-agencies/>) (2024).
21. The Clean Energy Transition Partnership. Full statement on international public support for the clean energy transition. (available at: <https://cleanenergytransitionpartnership.org/>) (2024).
22. International Energy Agency. Coal 2023 – Analysis and forecast to 2026. (available at: https://iea.blob.core.windows.net/assets/a72a7ffa-c5f2-4ed8-a2bf-eb035931d95c/Coal_2023.pdf) (2023).
23. Berne Union. Members. (available at: <https://www.berneunion.org/Members>) (2024).
24. Waidelich, P. & Steffen, B. Renewable energy financing by state investment banks: Evidence from OECD countries. *Energy Econ.* **132**, 107455 (2024).
25. Lonergan, K. E. *et al.* Improving the representation of cost of capital in energy system models. *Joule* **7**, 469–483 (2023).
26. Briera, T. & Lefèvre, J. Reducing the cost of capital through international climate finance to accelerate the renewable energy transition in developing countries. *Energy Policy* **188**, 114104 (2024).
27. Calcaterra M. *et al.* Reducing cost of capital to finance the energy transition in developing countries: a multi-model analysis. *Nat. Energy* **forthcoming** (2024).
28. International Energy Agency. World energy investment 2023 – analysis. (available at: <https://www.iea.org/reports/world-energy-investment-2023>) (2023).
29. Deutscher Bundestag. Antwort des Staatssekretärs Udo Philipp vom 27. März 2023 - Drucksache 20/6259 [Response to parliamentary inquiry]. (available in German at:

<https://dserver.bundestag.de/btd/20/062/2006259.pdf> (2023).

30. The World Bank Group. International Development Association. *IDA financing*. (available at: <https://ida.worldbank.org/en/financing>) (2024).

31. Dooley, K. *et al.* Ethical choices behind quantifications of fair contributions under the Paris Agreement. *Nat. Clim. Change* **11**, 300–305 (2021).

32. Naidoo, C. P. *et al.* Exploring the intersection of equity and Article 2.1(c) - Towards an improved global stocktake. *IGST Des. Robust Stock. Discuss. Ser.* (available at: <https://odi.org/en/publications/exploring-the-intersection-of-equity-and-article-21c-towards-an-improved-global-stocktake/>) (2023).

33. Mulugetta, Y. *et al.* Africa needs context-relevant evidence to shape its clean energy future. *Nat. Energy* **7**, 1015–1022 (2022).

34. Sanchez, F. & Linde, L. Turning out the light: criteria for determining the sequencing of countries phasing out oil extraction and the just transition implications. *Clim. Policy* **23**, 1182–1196 (2023).

35. Rasul, M. G., Hazrat, M. A., Sattar, M. A., Jahirul, M. I. & Shearer, M. J. The future of hydrogen: Challenges on production, storage and applications. *Energy Convers. Manag.* **272**, 116326 (2022).

36. TXF. NEOM Green hydrogen project (NGHP) - project financing. *Deal overview* (available at: <https://www.txfnews.com/data/deals/21962/neom-green-hydrogen-project-nghp-project-financing>) (2024).

37. Ueckerdt, F. *et al.* Potential and risks of hydrogen-based e-fuels in climate change mitigation. *Nat. Clim. Change* **11**, 384–393 (2021).

38. Devlin, A., Kossen, J., Goldie-Jones, H. & Yang, A. Global green hydrogen-based steel opportunities surrounding high quality renewable energy and iron ore deposits. *Nat. Commun.* **14**, 2578 (2023).

39. Ma, Y. *et al.* Reducing iron oxide with ammonia: A sustainable path to green steel. *Adv. Sci.* **10**, 2300111 (2023).

40. Keating, S. H2 Green Steel Boden: Long on ambitions, low on emissions. *TXF Features*. (available at: <https://www.txfnews.com/articles/7631/h2-green-steel-boden-long-on-ambitions-low-on-emissions>) (2024).

41. Garrett-Peltier, H. Green versus brown: Comparing the employment impacts of energy efficiency, renewable energy, and fossil fuels using an input-output model. *Econ. Model.* **61**, 439–447 (2017).

42. Molnár, B., Frizis, I., Van Hummelen, S. & Stenning, J. Export credit support in the Netherlands: fossil phase out and job impacts. (available at: https://www.bothends.org/uploaded_files/document/2022_NL_export_credits_jobs_study_.pdf) (2022).

43. Vivid Economics. UK Export Finance and domestic jobs. (available at:

http://www.vivideconomics.com/wpcontent/uploads/2020/10/20201012-UKEF-and-domestic-jobs_3rd-draft_clean.pdf) (2020).

44. Diluiso, F. *et al.* Coal transitions—part 1: a systematic map and review of case study learnings from regional, national, and local coal phase-out experiences. *Environ. Res. Lett.* **16**, 113003 (2021).

45. Nacke, L., Vinichenko, V., Cherp, A., Jakhmola, A. & Jewell, J. Compensating affected parties necessary for rapid coal phase-out but expensive if extended to major emitters. *Nat. Commun.* **15**, 3742 (2024).

46. Jensen, L. Global decarbonization in fossil fuel export-dependent economies - Fiscal and economic transition costs. *Dev. Future Ser.* (available at: <https://www.undp.org/publications/dfs-global-decarbonization-fossil-fuel-export-dependent-economies>) (2023).

47. Mercure, J.-F. *et al.* Macroeconomic impact of stranded fossil fuel assets. *Nat. Clim. Change* **8**, 588–593 (2018).

48. International Energy Agency. The oil and gas industry in net zero transitions – analysis. (available at: <https://www.iea.org/news/oil-and-gas-industry-faces-moment-of-truth-and-opportunity-to-adapt-as-clean-energy-transitions-advance>) (2023).

49. Welsby, D., Price, J., Pye, S. & Ekins, P. Unextractable fossil fuels in a 1.5 °C world. *Nature* **597**, 230–234 (2021).

50. Green, F., Kursk, O. B. von, Muttitt, G. & Pye, S. No new fossil fuel projects: The norm we need. *Science* **384**, 954-957 (2024).

51. Gallagher, K. P., Kamal, R., Jin, J., Chen, Y. & Ma, X. Energizing development finance? The benefits and risks of China’s development finance in the global energy sector. *Energy Policy* **122**, 313–321 (2018).

52. Kong, B. & Gallagher, K. P. The new coal champion of the world: The political economy of Chinese overseas development finance for coal-fired power plants. *Energy Policy* **155**, 112334 (2021).

53. U.S. Department of the Treasury. Joint statement on the temporary suspension of the technical negotiations in the International Working Group on Export Credits. *Statements & Remarks.* (available at: <https://home.treasury.gov/news/press-releases/sm1188>) (2020).

54. African Development Bank. Mozambique - enabling large scale gas and power investments in Mozambique technical assistance project - appraisal report. (available at: <https://www.afdb.org/en/documents/document/mozambique-enabling-large-scale-gas-and-power-investments-in-mozambique-technical-assistance-project-appraisal-report-34910>) (2013).

55. Eurostat. Inland demand of natural gas, EU, 1990-2023. (available at: [https://ec.europa.eu/eurostat/statistics-explained/index.php?title=File:F1-Inland_demand_of_natural_gas,_EU,_1990-2023_\(terajoules_\(Gross_Calorific_Value\)\).png](https://ec.europa.eu/eurostat/statistics-explained/index.php?title=File:F1-Inland_demand_of_natural_gas,_EU,_1990-2023_(terajoules_(Gross_Calorific_Value)).png)) (2024).

56. The World Bank Group. World development indicators - Mozambique. (available at: <https://data.worldbank.org/country/mozambique>) (2024).
57. Cotterill, J., White, S. & Wilson, T. Total accused of involuntary manslaughter over 2021 Mozambique attack. *Financial Times*. (available at: <https://www.ft.com/content/2c6812ad-9619-4a64-bfe2-ffcc8934f8d8>) (2023).
58. Thompson, M. Mozambique LNG: When will funds be disbursed? *TXF Features*. (Available at: <https://www.txfnews.com/articles/7671/mozambique-lng-when-will-funds-be-disbursed>) (2024).
59. International Energy Agency and International Finance Corporation. Scaling up private finance for clean energy in emerging and developing economies. (available at: <https://www.ifc.org/en/insights-reports/2023/scaling-up-private-finance-for-clean-energy-in-emdes>) (2023).
60. Thorbecke, W. Japanese economic performance after the pandemic: A sectoral analysis. *J. Risk Financ. Manag.* **16**, 267 (2023).